# A role of OCRL in clathrin-coated pit dynamics and uncoating revealed by studies of Lowe syndrome cells

Ramiro Nández[1,2†], Daniel M Balkin[1,2†], Mirko Messa[1,2], Liang Liang[3], Summer Paradise[1,2], Heather Czapla[1,2], Marco Y Hein[4], James S Duncan[3,5,6], Matthias Mann[4], Pietro De Camilli[1,2*]

[1]Department of Cell Biology, Howard Hughes Medical Institute, Yale University School of Medicine, New Haven, United States; [2]Program in Cellular Neuroscience, Neurodegeneration and Repair, Yale University School of Medicine, New Haven, United States; [3]Department of Diagnostic Radiology, Yale University School of Medicine, New Haven, United States; [4]Department of Proteomics and Signal Transduction, Max Planck Institute of Biochemistry, Martinsried, Germany; [5]Department of Biomedical Engineering, Yale University School of Medicine, New Haven, United States; [6]Department of Electrical Engineering, Yale University School of Medicine, New Haven, United States

*For correspondence: pietro.
decamilli@yale.edu

†These authors contributed
equally to this work

Competing interests: The
authors declare that no
competing interests exist.

Reviewing editor: Suzanne R
Pfeffer, Stanford University,
United States

**Abstract** Mutations in the inositol 5-phosphatase OCRL cause Lowe syndrome and Dent's disease. Although OCRL, a direct clathrin interactor, is recruited to late-stage clathrin-coated pits, clinical manifestations have been primarily attributed to intracellular sorting defects. Here we show that OCRL loss in Lowe syndrome patient fibroblasts impacts clathrin-mediated endocytosis and results in an endocytic defect. These cells exhibit an accumulation of clathrin-coated vesicles and an increase in U-shaped clathrin-coated pits, which may result from sequestration of coat components on uncoated vesicles. Endocytic vesicles that fail to lose their coat nucleate the majority of the numerous actin comets present in patient cells. SNX9, an adaptor that couples late-stage endocytic coated pits to actin polymerization and which we found to bind OCRL directly, remains associated with such vesicles. These results indicate that OCRL acts as an uncoating factor and that defects in clathrin-mediated endocytosis likely contribute to pathology in patients with OCRL mutations.

## Introduction

Reversible phosphorylation of the inositol ring of phosphatidylinositol at the 3, 4 and 5 position generates seven phosphoinositide species, which play critical regulatory roles in cell physiology via their property to control interactions at the cytosolic surface of membranes. Phosphoinositide levels are dynamically and spatially regulated by kinases and phosphatases, establishing a code of membrane identity (*Di Paolo and De Camilli, 2006*; *Vicinanza et al., 2008*; *Balla, 2013*). The human genome contains 10 genes encoding inositol 5-phosphatases, a group of enzymes that dephosphorylate the inositol ring at the 5 position, nine of which act on inositol phospholipids (*Dyson et al., 2012*; *Pirruccello and De Camilli, 2012*). Mutations in one such gene, OCRL, give rise to Oculo-Cerebro-Renal syndrome of Lowe (Lowe syndrome) and type 2 Dent's disease, two X-linked diseases (*Attree et al., 1992*; *Hoopes et al., 2005*). Lowe syndrome is characterized by congenital cataracts, renal proximal tubule dysfunction, cognitive disabilities and developmental delay (*Delleman et al., 1977*; *Kenworthy and Charnas, 1995*; *Böckenhauer et al., 2008*). Patients with Dent's disease exhibit similar proximal tubule defects but no, or only mild, additional clinical defects (*Bökenkamp et al., 2009*; *Shrimpton et al., 2009*).

**eLife digest** Oculo-Cerebro-Renal syndrome of Lowe (Lowe syndrome) is a rare genetic disorder that can cause cataracts, mental disabilities and kidney dysfunction. It is caused by mutations in the gene encoding OCRL, a protein that modifies a membrane lipid and that is found on membranes transporting molecules (cargo) into cells by a process known as endocytosis.

During endocytosis, the cell outer membrane is deformed into a pit that engulfs the cargo to be taken up by the cell. The pit then pinches off from the outer membrane to form a vesicle—a bubble-like compartment—inside the cell that transports the cargo to its destination. In one type of endocytosis, this process is mediated by a basket-like coat primarily made up from the protein clathrin that assembles at the membrane patch to be internalized. After the vesicle is released from the cell membrane, the clathrin coat is broken apart and its components are shed and recycled for use by new budding endocytic vesicles.

The OCRL protein had previously been observed associated to newly forming clathrin-coated vesicles, but the significance of this was not known. Now, Nández et al. have used a range of imaging and analytical techniques to further investigate the properties of OCRL, taking advantage of cells from patients with Lowe syndrome. These cells lack OCRL, and so allow the effect of OCRL's absence on cell function to be deduced. OCRL destroys the membrane lipid that helps to connect the clathrin coat to the membrane, and Nández et al. show that without OCRL the newly formed vesicle moves into the cell but fails to efficiently shed its clathrin coat. Thus, a large fraction of clathrin coat components remain trapped on the vesicles, reducing the amount of such components available to help new pits develop into vesicles. As a consequence, the cell has difficulty internalizing molecules.

Collectively, the findings of Nández et al. outline that OCRL plays a role in the regulation of endocytosis in addition to its previously reported actions in the control of intracellular membrane traffic. The results also help to explain some of the symptoms seen in Lowe syndrome patients.

OCRL has a modular structure that is comprised of an N-terminal PH domain (*Mao et al., 2009*) followed in sequence by an unstructured amino acid stretch, the 5-phosphatase domain, an ASPM-SPD2-Hydin (ASH) domain and a catalytically-inactive RhoGAP-like domain (*Ponting, 2006*; *Erdmann et al., 2007*). OCRL is expressed in all tissues and has a broad subcellular distribution, which is mediated by interactions with proteins that control membrane traffic and the actin cytoskeleton. These include clathrin-heavy chain, the clathrin adaptor AP-2, Rab and Rho family GTPases and the endosomal adaptors APPL1 and Ses1/2 (*Ungewickell et al., 2004*; *Choudhury et al., 2005, 2009*; *Shin et al., 2005*; *Hyvola et al., 2006*; *Lichter-Konecki et al., 2006*; *Erdmann et al., 2007*; *Fukuda et al., 2008*; *Swan et al., 2010*; *Dambournet et al., 2011*; *Noakes et al., 2011*; *Pirruccello et al., 2011*). Consistent with such interactions, OCRL has been localized to the Golgi complex, to a variety of endosomal compartments as well as to clathrin-coated pits (*Olivos-Glander et al., 1995*; *Dressman et al., 2000*; *Ungewickell et al., 2004*; *Choudhury et al., 2005*; *Faucherre et al., 2005*; *Hyvola et al., 2006*; *Erdmann et al., 2007*). INPP5B, a 5-phosphatase very similar to OCRL, has many of the same properties, but does not bind clathrin and AP-2, and correspondingly is not localized at endocytic clathrin-coated pits (*Erdmann et al., 2007*; *Williams et al., 2007*).

PI(4,5)P$_2$ and PI(3,4,5)P$_3$, the two preferred substrates of OCRL, are primarily concentrated in the plasma membrane, where they help define its identity. The broad intracellular localization of OCRL may reflect its importance in preventing the build-up of ectopic intracellular pools of PI(4,5)P$_2$ and PI(3,4,5)P$_3$. Alternatively, OCRL may act on small yet physiologically relevant pools of these phosphoinositides on intracellular membranes. The pathological manifestations of Lowe syndrome and Dent's disease are thought to arise from abnormal protein sorting and membrane recycling (*Erdmann et al., 2007*; *Swan et al., 2010*; *Vicinanza et al., 2011*; *Mehta et al., 2014*), as loss of OCRL would disrupt the appropriate coordination between membrane progression and changes in phosphoinositide composition.

One open question is the physiological significance of the localization of OCRL at endocytic clathrin-coated pits. Nucleation and expansion of endocytic clathrin coats requires PI(4,5)P$_2$ at the plasma membrane. PI(4,5)P$_2$ represents a critical co-receptor for all the clathrin adaptors (*Di Paolo and De Camilli, 2006*; *Mettlen et al., 2009*; *Sun et al., 2007*; *Traub, 2009*) and its levels (controlled by dynamic turnover)

impact the lifetime of clathrin-coated pits (*Zoncu et al., 2007*; *Nakatsu et al., 2010*; *Antonescu et al., 2011*). Conversely, coat shedding (uncoating) after the endocytic reaction requires PI(4,5)P$_2$ dephosphorylation, as first revealed by studies of clathrin-mediated endocytosis in axon terminals (*Cremona et al., 1999*). Absence of synaptojanin 1, the most abundant 5-phosphatase in the synapse, results in a striking accumulation of clathrin-coated vesicles (*Cremona et al., 1999*; *Harris et al., 2000*; *Verstreken et al., 2003*; *Milosevic et al., 2011*). There is also evidence that PI(4,5)P$_2$ dephosphorylation may assist in the fission reaction of endocytosis (*Stefan et al., 2002*; *Rusk et al., 2003*; *Mani et al., 2007*; *Sun et al., 2007*; *Liu et al., 2009*; *Chang-Ileto et al., 2011*). While it has been shown that OCRL has a role in preventing the accumulation of PI(4,5)P$_2$ on post-endocytic membranes (*Vicinanza et al., 2011*), there is no evidence to date that the absence of OCRL, or defects in OCRL function, impairs clathrin-mediated endocytosis.

The initial goal of this study was to gain new insights into the function of OCRL through a global analysis of the OCRL interactome. Such an analysis corroborated evidence for a primary role of OCRL at endocytic clathrin-coated pits as it confirmed clathrin and other endocytic adaptors as OCRL binding partners. In addition, it revealed that SNX9, a clathrin-binding protein associated with late-stage endocytic clathrin-coated pits, directly binds OCRL. Prompted by these findings, we examined the contribution of OCRL to clathrin-mediated endocytosis in Lowe syndrome patient fibroblasts. We found that the absence of OCRL in patient cells impairs endocytic clathrin coat dynamics and results in an endocytic defect. These cells accumulate clathrin-coated vesicles that fail to loose their coat. Such vesicles nucleate at least a cohort of the intracellular actin comets previously described in Lowe syndrome patient cells (*Suchy and Nussbaum, 2002*; *Allen, 2003*; *Hayes et al., 2009*; *Cui et al., 2010*; *Dambournet et al., 2011*; *Vicinanza et al., 2011*). Our findings suggest that OCRL participates in uncoating and thus facilitates the recycling of endocytic factors, a process required for the normal progression of endocytosis. Defects in clathrin-mediated endocytosis resulting from the lack of OCRL may play a role in the clinical manifestations of Lowe syndrome and Dent's disease.

## Results

### OCRL interactome reveals a major link to clathrin-mediated traffic

The binding partners of an enzyme define its sites of action and regulatory mechanisms. Towards an improved understanding of the function of OCRL, we carried out a global analysis of its protein interactome. OCRL immunoprecipitates were analyzed through quantitative label-free interaction proteomics by mass spectrometry. This technique, in contrast to other methods such as yeast two-hybrids or pull-downs with purified proteins, allows the preferential capture of physiologically occurring protein complexes (*Vermeulen et al., 2008*). To this aim, HeLa M cell lines stably expressing either GFP-OCRL or GFP alone were generated (*Figure 1A–C*) so that OCRL specific interactions could be identified by analysis of proteins selectively recovered in anti-GFP immunoprecipitates from cells expressing GFP-OCRL. While GFP alone had a diffuse cytosolic distribution, GFP-OCRL had a punctate localization characteristic of endogeneous OCRL immunoreactivity (*Figure 1A–C*), thus validating GFP-OCRL expressing cells as an appropriate model for our experiments. Two cell lines were used for further analysis, which expressed GFP-OCRL at near endogenous levels (1X) or at about five times the endogenous level (5X), as revealed by western blotting against OCRL (*Figure 1D*).

Inspection of anti-GFP immunoprecipitates generated from these cells by SDS-PAGE gels prior to mass spectrometry revealed a very robust specific enrichment of GFP-OCRL in samples from GFP-OCRL expressing cells (*Figure 1E*). As expected, a major band was detected at 170 KD (*Figure 1E*), shown to be clathrin heavy chain by western blotting (*Figure 1F*; *Ungewickell et al., 2004*; *Choudhury et al., 2005*). In addition, there were numerous other bands selectively observed in GFP-OCRL samples. Mass spectrometry further confirmed clathrin heavy and light chain as major hits in the immunoprecipitates and revealed additional specific OCRL interactors (*Figure 1—figure supplement 1*; *Supplementary file 1A,B*). These included known binding partners of OCRL, such as subunits of the endocytic clathrin adaptor AP-2, Rab proteins and Ses1/2 but, surprisingly, not APPL1. Possibly APPL1 was outcompeted by Ses1/2, which bind to the same site in OCRL but with a higher affinity (*Swan et al., 2010*; *Noakes et al., 2011*; *Pirruccello et al., 2011*). Specific interactors also included numerous proteins implicated in membrane trafficking, primarily along the endocytic pathway and vesicular transport steps between endosomes and the Golgi complex (*Supplementary file 1A,B*). Proteins that participate in clathrin-dependent transport steps featured particularly prominently in the OCRL interactome.

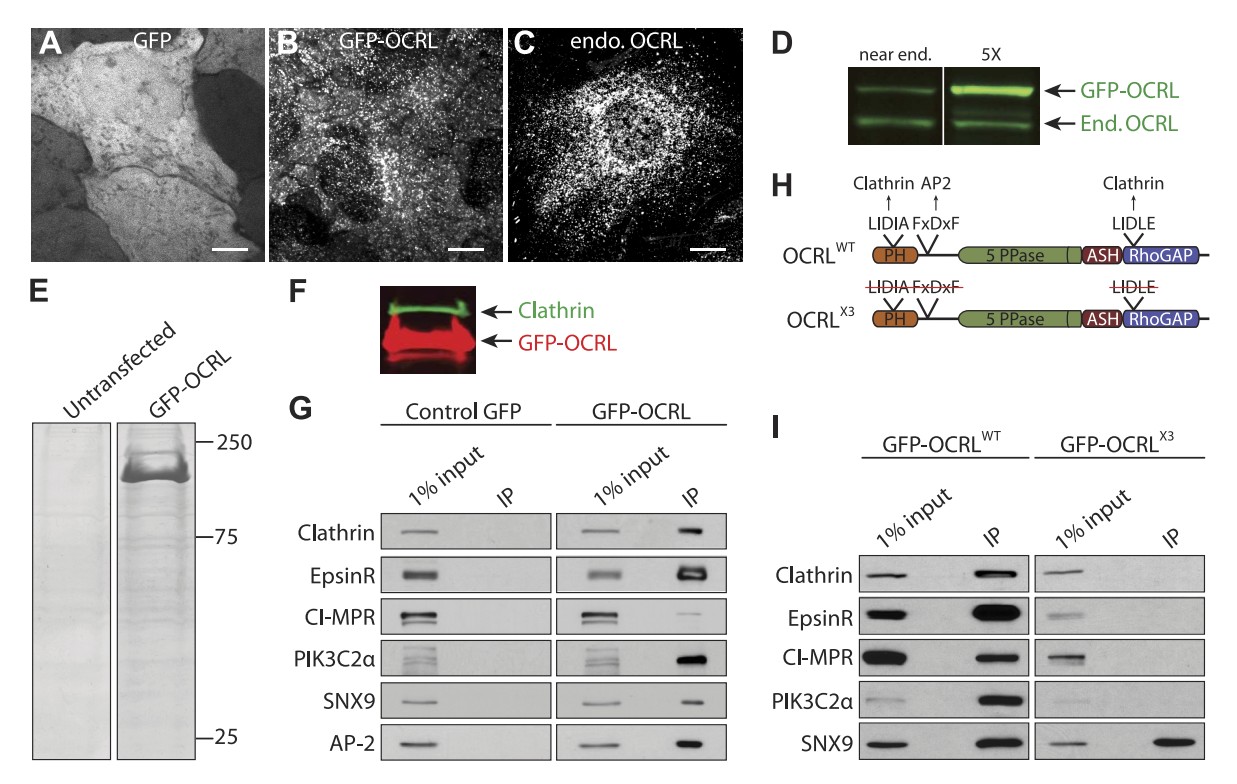

**Figure 1**. The OCRL interactome strongly links this protein to clathrin-dependent membrane traffic. (**A** and **B**) GFP fluorescence in HeLa M cells stably expressing GFP alone (**A**) or GFP-OCRL (**B**). Scale bar: 10 µm. (**C**) Endogenous OCRL immunoreactivity in HeLa M cells. Scale bar: 10 µm. (**D**) Anti-OCRL western blot of HeLa M cells stably expressing GFP-OCRL at near endogenous (End.) levels (left) and at five times (5X) the endogenous levels (right). (**E**) Coomasie Blue stained SDS-PAGE of proteins immunoprecipitated from control and GFP-OCRL stably expressing HeLa M cells using magnetic beads coupled to anti-GFP antibody. (**F**) Immunoblot analysis of the material shown in (**E**) with antibodies directed against OCRL (red) and clathrin (green), revealing the strong recovery of clathrin along with OCRL in the anti-GFP immunoprecipitates (IP). (**G**) Western blot analysis for OCRL interactors identified by mass spectrometry in anti-GFP immunoprecipitates from cells expressing GFP alone or GFP-OCRL. (**H**) Domain structure of OCRL indicating binding motifs for clathrin and AP-2. In the OCRL$^{X3}$ mutant all three binding motifs are abolished (***Mao et al., 2009***). (**I**) Comparative immunoprecipitation from HeLa M cells expressing GFP-OCRL$^{WT}$ or GFP-OCRL$^{X3}$, followed by immunoblot analysis for the indicated proteins.

The following figure supplement is available for figure 1:

**Figure supplement 1**. OCRL interactors identified by label-free quantitative proteomics in immunoprecipitates generated from cell lines expressing GFP-OCRL approximately at endogenous levels (1X) and 5-times (5X) higher levels.

The enrichment of these proteins in the GFP-OCRL immunoprecipitate relative to the homogenate was further assessed by western blot analysis. In addition to clathrin, interactors such as AP-2, EpsinR (CLINT1), SNX9 and PI3KcIIα showed the highest enrichment by western blotting (***Figure 1G***). The cation-independent mannose 6-phosphate receptor (CI-M6PR), a cargo protein for clathrin-coated pits whose transport was previously shown to be affected by OCRL knockdown (***Vicinanza et al., 2011***), was also confirmed by western blotting, but was not as enriched (***Figure 1G***). Clathrin and AP-2 are known to bind OCRL directly, while EpsinR, SNX9 and PI3KcIIα are all known clathrin interactors (***Gaidarov et al., 2001***; ***Lundmark and Carlsson, 2003***; ***Mills et al., 2003***). These findings raised the possibility that at least some of these proteins may bind OCRL indirectly through clathrin. In fact, a substantial overlap was observed between major OCRL interactors revealed by our experiments and proteins previously identified in a similar mass spectrometry analysis of clathrin binding partners (***Hubner et al., 2010***) as well as in a global analysis of clathrin-coated vesicles (***Blondeau et al., 2004***; ***Borner et al., 2006***). Thus, to determine the contribution of clathrin to the interactions of OCRL, anti-GFP immunoprecipitations were repeated with cells expressing a GFP-OCRL construct lacking clathrin-binding sites.

## SNX9 is a novel direct OCRL interactor

Anti-GFP immunoprecipitions were performed on extracts from cells expressing either GFP-OCRL[WT] (wild type OCRL) or GFP-OCRL[X3] (an OCRL mutant in which both clathrin boxes as well as the AP-2-binding motif are mutated [*Mao et al., 2009*]) (*Figure 1H*). Western blot analysis revealed that loss of clathrin and AP-2 binding was accompanied by the loss of CI-M6PR, EpsinR and PI3KcIIα in the anti-GFP-OCRL[X3] immunoprecipitates (*Figure 1I*). In contrast, the interaction of OCRL with SNX9, which also contains binding sites for clathrin and for AP-2 (*Lundmark and Carlsson, 2003*), was not abolished by the three mutations in OCRL (*Figure 1I*). This prompted us to explore the possibility of a direct interaction between OCRL and SNX9.

SNX9 binds other proteins implicated in the late stages of clathrin-mediated endocytosis, such as dynamin, synaptojanin and N-WASP, through an N-terminal SH3 domain that recognizes proline-rich motifs with the canonical PxxP sequence (*Figure 2A*; *Badour et al., 2007*; *Lundmark and Carlsson, 2003*; *Mayer, 2001*; *Yeow-Fong et al., 2005*). OCRL contains a PxxP site ([175]REPPPPP[181]) in the predicted unfolded region that connects its N-terminal PH domain to its central inositol 5-phosphatase domain (*Figure 2A*). Anti-GFP immunoprecipitation of cell extracts expressing GFP-OCRL[WT] co-enriched endogenous SNX9, while a GFP-OCRL mutant in which three prolines (P177, 178, and 181) of the PxxP motif were mutated to alanine (GFP-OCRL[P3]), failed to co-precipitate SNX9 (*Figure 2B*). Conversely, GFP-SNX9[WT] but not a GFP-SNX9 construct lacking the SH3 domain (GFP-SNX9[ΔSH3]) co-precipitated endogenous OCRL (*Figure 2C*). Therefore, we concluded that the PxxP motif in OCRL is necessary for the interaction with SNX9. A close paralogue of SNX9 with partially overlapping functions is SNX18 (*Park et al., 2010*). Anti-GFP immunoprecipitations from cells expressing a GFP fusion of SNX18 also recovered endogenous OCRL (*Figure 2—figure supplement 1*).

To determine whether the PxxP motif of OCRL was sufficient to bind SNX9, GST fusions of N-terminal fragments of OCRL comprising the PH domain and portions of its flanking unfolded region, including (GST-OCRL[217]) or excluding (GST-OCRL[176]) the PxxP motif, were tested in pull-down experiments from mouse brain lysates (*Figure 2D*). Both fragments contained a previously described clathrin box ([73]LIDIA[77]) (*Mao et al., 2009*) and each recovered clathrin, as expected (*Figure 2D*). However, only the OCRL[217] fragment containing the PxxP motif pulled down SNX9 (*Figure 2D*). Collectively, these results indicate that SNX9 directly binds OCRL via an interaction of its SH3 domain with the PxxP site in OCRL.

## Sequential recruitment of SNX9 and OCRL at endocytic clathrin-coated pits

A direct binding of SNX9 to OCRL is consistent with the localization of both proteins at late-stage endocytic clathrin-coated pits, as shown by the analysis of these two proteins expressed independently (*Soulet et al., 2005*; *Erdmann et al., 2007*; *Yarar et al., 2007*; *Taylor et al., 2011*). Based on a systematic analysis of the peak of endocytic protein recruitment at pits relative to the closure of their neck, *Taylor et al. (2011)* further reported that OCRL peaks slightly after SNX9. To directly compare the spatiotemporal relationship between two proteins at clathrin-coated pits, time-lapse confocal microscopy was performed on human wild type fibroblasts co-expressing GFP-SNX9 and mCherry-OCRL. Inspection of these cells revealed rapidly turning over puncta of both OCRL and SNX9 uniformly distributed on the plasma membrane, as expected for late stage clathrin-coated pits (*Figure 2E*). In the great majority of cases (93.18 ± 1.25%; n = 150 events, 3 independent experiments), the peak of OCRL fluorescence lagged slightly behind that of SNX9, suggesting a very late function of OCRL in clathrin-mediated budding (*Figure 2E–G*; *Video 1*).

These findings support a role of OCRL in the very late stages of clathrin-mediated endocytosis. However, evidence from loss-of-function studies confirming this role is still missing. To investigate this possibility, we examined endocytic clathrin-coated pits and their dynamics in patient cells that lack OCRL.

## Abnormal distribution of clathrin coat components in Lowe syndrome fibroblasts

Dermal fibroblasts derived from a Lowe syndrome patient and from a control subject were used for these studies. Patient cells lack OCRL, as indeed confirmed by immunofluorescence (*Figure 3A,B*) and western blotting (*Figure 3C*). Accordingly, a near 1.5-fold increase in the steady state levels of PI(4,5)$P_2$ was also observed in these cells as expected (*Figure 3D*; *Zhang et al., 1998*; *Wenk et al., 2003*).

Immunofluorescence for endogenous clathrin and AP-2 showed a robust increase of their punctate immunoreactivity in patient cells (*Figure 3E–H*). An even more striking increase was observed for

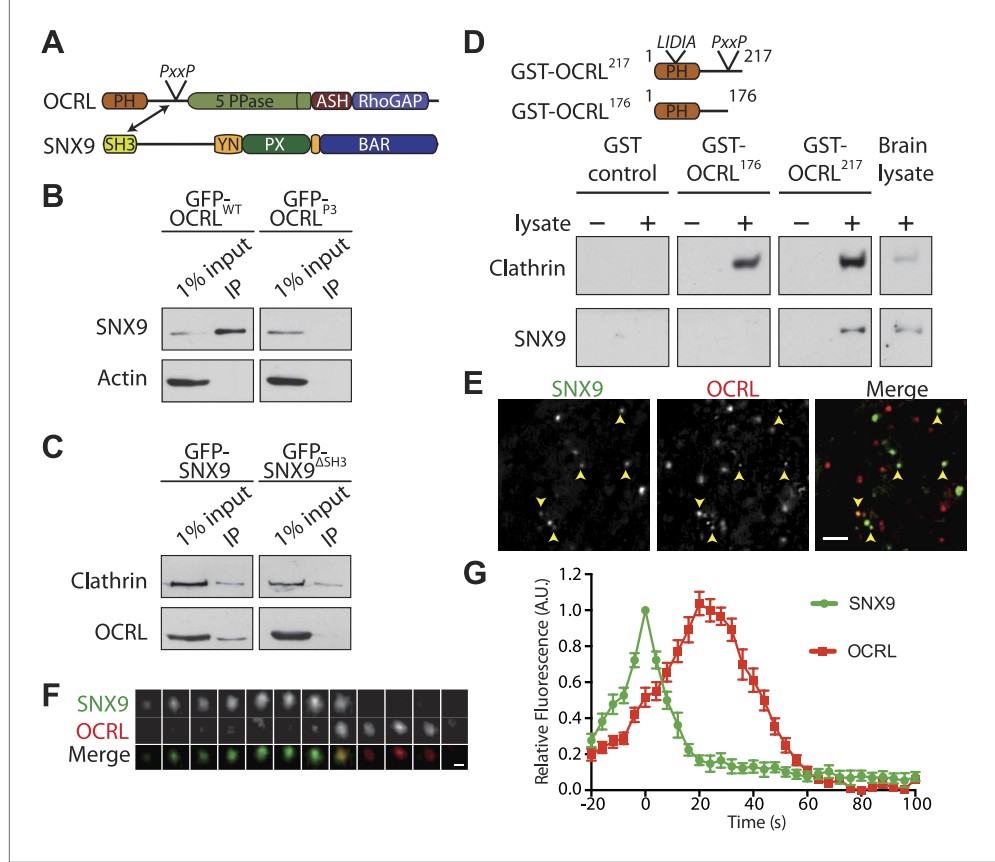

**Figure 2**. SNX9 is a novel direct interactor of OCRL at clathrin-coated pits. (**A**) Domain structure of OCRL and SNX9. The location of the PxxP site in OCRL is indicated. (**B**) Western blot showing the recovery of SNX9 in immunoprecipitates generated from HEK293T cells expressing full length GFP-OCRL[WT] or a GFP-OCRL mutant in which the PxxP site was mutated (GFP-OCRL[P3]). Endogenous SNX9 was recovered only in the immunoprecipitates of GFP-OCRL[WT]. Actin is shown as a control. (**C**) Western blot showing the recovery of endogenous OCRL in immunoprecipitates from cells expressing GFP-SNX9[WT] but not from cells expressing GFP-SNX9 mutant missing the SH3 domain (GFP-SNX9[ΔSH3]). Note that binding of SNX9 to clathrin, which binds SNX9 outside the SH3 domain, is not lost. (**D**) (Top) Schematic representation of GST-OCRL constructs comprising the PH domain and portions of the predicted unstructured flanking region including (GST-OCRL[217]) or excluding (GST-OCRL[176]) the PxxP motif. Both constructs include the clathrin-box found inside the PH domain. (Bottom) GST pulldowns from a mouse brain extract using GST alone or GST fusions of OCRL as baits. Subsequent western blot analysis of the bound material shows that both constructs recover clathrin, but only GST-OCRL[217] recovers SNX9. (**E**) Confocal microscopy of wild type human fibroblasts expressing GFP-SNX9 and mCherry-OCRL. Most SNX9 spots colocalize with OCRL (yellow arrowheads) and nearly all SNX9 spots acquire OCRL during the acquisition. Scale bar: 2 μm. (**F**) Sequential confocal images (8 s apart) of an SNX9 spot that acquires OCRL. Scale bar: 0.3 μm. (**G**) Average time course of fluorescence intensity for SNX9 and OCRL at individual pits. The peak of SNX9 fluorescence was defined as time zero (mean ± SEM, n = 150 events, 10 different cells).
The following figure supplement is available for figure 2:

**Figure supplement 1**. SNX18 interacts with OCRL.

SNX9 puncta (*Figure 3I–J*) (see also below). Additionally, clathrin-coated pits showed a clustered distribution. Exogenous overexpression of GFP-OCRL[WT] rescued this phenotype (*Figure 3K–K'*), while expression of a catalytically inactive OCRL (GFP-OCRL[D523G]) failed to do so (*Figure 3L–L'*). Such phenotypes were reminiscent of the accumulation of puncta positive for endocytic clathrin coat components observed in cells where the progression of late stage clathrin-coated pits to free vesicles is impaired (*Ferguson et al., 2009*; *Posor et al., 2013*), strongly suggesting that lack of OCRL affects the completion of the endocytic reaction. Thus far, defects in clathrin-mediated endocytosis have not

**Video 1**. Sequential recruitment of SNX9 and OCRL at clathrin-coated pits. Wild-type skin fibroblast co-transfected with mCherry-OCRL and GFP-SNX9 showing the punctate localization of SNX9 and OCRL at clathrin-coated pits and the sequential recruitment of SNX9 followed by OCRL at the pit.

been reported in Lowe syndrome fibroblasts, and therefore we next analyzed clathrin-coated pit dynamics in patient and control cells by time-lapse microscopy.

## Decreased turnover of endocytic clathrin-coated pits and delayed endocytosis in patient cells

The dynamics of endocytic clathrin-coated pits in human fibroblasts were assessed by monitoring the fluorescence of GFP-tagged μ2 subunit of the clathrin adaptor AP-2 (μ2-GFP) by live Total Internal Reflection Fluorescence (TIRF) microscopy, followed by an automated analysis of μ2 puncta to determine their global lifetime distributions (*Liang et al., 2014*). Using expectation-maximization algorithms, a Gaussian mixture model was then fitted to each population of puncta, grouping events into three distinct cohorts: short, medium and long with mean lifetimes of 40 s (short), 108 s (medium) and 244 s (long), respectively. As expected, μ2 puncta exhibited dynamic turnover in control cells, with only a small fraction of long-lived pits (short: 36.8%, medium: 46.6%, long: 16.6%) (*Figure 4A*). In patient cells, the turnover of clathrin-coated pits was reduced, as revealed by a significant increase in the number of long-lived pits and a reduction of short-lived ones (short: 22.3%, medium: 42.3%, long: 35.4%) (*Figure 4B*). This phenotype was rescued by the re-expression of wild type OCRL (short: 33.8%, medium: 49.9%, long: 16.3%) (*Figure 4C*).

Confocal microscopy was also used to examine control and patient cells transfected with mRFP-tagged clathrin light chain (mRFP-CLC) and either wild type or a catalytically inactive GFP-tagged OCRL (GFP-OCRL[WT] or GFP-OCRL[D523G]). In control cells, GFP-OCRL[WT] was recruited to clathrin spots as the clathrin fluorescence started to diminish (*Figure 4D*; *Video 2*), in agreement with previous observations (*Erdmann et al., 2007*). The signal for OCRL then rapidly disappeared as the free vesicle moved away from the imaging plane. In patient cells, the dynamics of the pits were disrupted, as demonstrated by the longer persistence of the clathrin signal (*Figure 4E*; *Video 3*). GFP-OCRL[WT] restored normal clathrin turnover in these cells (*Figure 4F*; *Video 4*), while GFP -OCRL[D523G] failed to do so despite being recruited to the pits and remaining associated with them (*Figure 4E*; *Video 3*). Delayed turnover in patient cells resulted in an increase in the number of clathrin puncta (mRFP-tagged CLC) that were positive for GFP-OCRL[D523G] at any given time (*Figure 4G*).

To assess the stage at which endocytic clathrin-coated pits were stalled in patient fibroblasts, electron microscopy was performed (*Figure 4H*). This analysis confirmed the global increase in the number of clathrin-coated endocytic profiles and, in particular, a prominent increase of wide-neck (primarily U-shaped) pits and of free clathrin-coated vesicles in the cortical region of the cell (less than 500 nm from the plasma membrane, that is clathrin-coated vesicles most likely of endocytic origin) (*Figure 4H*). These results suggest that the loss of OCRL impairs uncoating, a finding consistent with its very late recruitment at pits. Its absence also delays late steps in clathrin-mediated endocytosis, possibly as a result of sequestration of endocytic factors on uncoated vesicles. Such findings prompted us to assess the potential occurrence in patient cells of a defect in the internalization of transferrin, a well established cargo of clathrin-mediated endocytosis.

Confocal microscopy of cells pre-incubated with fluorescent transferrin revealed a delay in the shift of fluorescence from the cell surface to deeper positions in the cytoplasm in patient cells compared to control fibroblasts (*Figure 4I–J*). This defect was quantified and confirmed biochemically by measuring the uptake of biotinylated transferrin at various time points during a 20-min time course (*Figure 4K*). In addition, biotin labeling of the endogenous surface pool of the transferrin receptor showed its accumulation at the plasma membrane in patient cells compared to controls (*Figure 4L*). From these results, we conclude OCRL is required for normal dynamics of clathrin-mediated endocytosis.

## Actin comets in Lowe syndrome patient cells nucleated by endocytic vesicles

A reported phenotype of Lowe syndrome patient fibroblasts is an abnormal organization of actin, with the presence of numerous actin comets (*Suchy and Nussbaum, 2002*; *Allen, 2003*; *Hayes et al., 2009*;

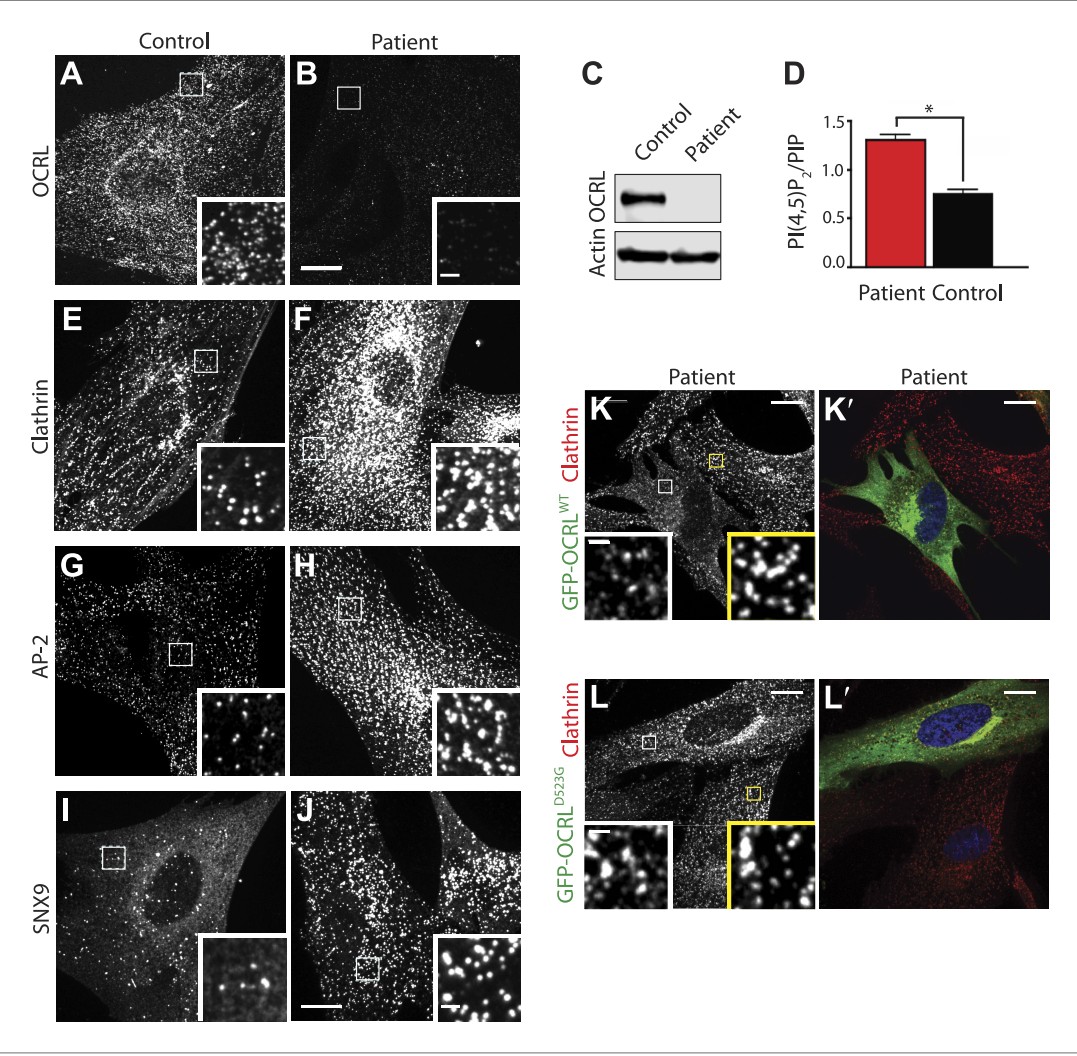

**Figure 3**. Abnormal distribution of endocytic factors in Lowe syndrome fibroblasts. (**A** and **B**) Immunofluorescence staining for OCRL in control (**A**) and Lowe syndrome fibroblasts (**B**), revealing the lack of OCRL in patient cells. Insets show higher magnifications of the boxed regions. Scale bar full size: 10 µm; inset: 1 µm. (**C**) Immunoblot analysis using a monoclonal antibody directed against the C-terminal region of human OCRL demonstrates the absence of the protein in patient cells. Actin is shown as a loading control. (**D**) High pressure liquid chromatography (HPLC) analysis of cell extracts showing an increase in PI(4,5)P$_2$ levels in patient cells (Student's *t* test, *p = 0.01). (**E**–**J**) Immunofluorescence staining of control (left) and Lowe syndrome fibroblasts (right) for clathrin (**E**–**F**), AP-2 (**G**–**H**) and SNX9 (**I**–**J**) revealing the enhanced punctate immunoreactivity for endocytic factors in patient cells. Insets show higher magnifications of the boxed regions. Scale bar full size: 10 µm; inset: 1 µm. (**K**–**L′**) Rescue of the clathrin phenotype (**K**) and (**L**) in patient cells following expression of GFP-OCRL[WT] (**K′**) but not in cells expressing the catalytically inactive GFP-OCRL[D523G] (**L′**). Insets show higher magnifications of the boxed regions. Region and inset framed in white depict cells expressing the transfected construct (either GFP-OCRL[WT] (**K**–**K′**) or OCRL[D523G] (**L**–**L′**)). Region and inset framed in yellow show cells that did not express the GFP-tagged constructs. Scale bar full size: 10 µm; inset: 1 µm.

*Cui et al., 2010*; *Dambournet et al., 2011*; *Vicinanza et al., 2011*). Accordingly, analysis of F-actin in control and patient cells using either phalloidin (in fixed cells) or mCherry-CH[UTR] (in live cells) (*Burkel et al., 2007*) revealed a reduction in the number of actin stress fibers, with the prominent accumulation of peripheral actin foci and actin comets (also referred to as actin tails) in patient cells (*Figure 5A–B*; *Video 5*). These comets, which are dependent on N-WASP and the Arp2/3 complex (*Figure 5C*, *Figure 5—figure supplement 1A–F*), are thought to reflect the nucleation of actin on intracellular vesicles that ectopically accumulate PI(4,5)P$_2$ due to the lack of OCRL (*Vicinanza et al., 2011*). Consistent with

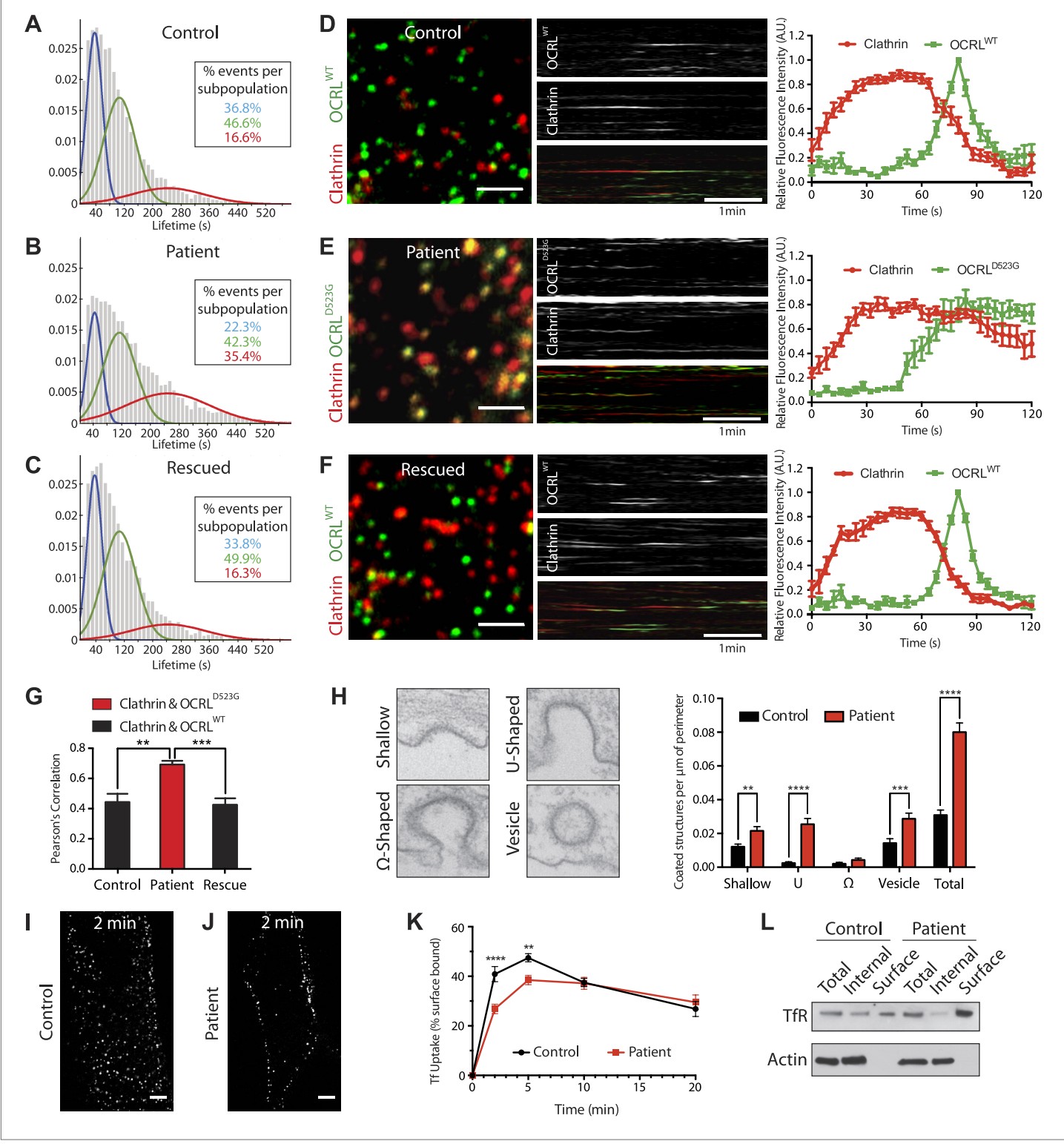

**Figure 4.** Defects in clathrin-mediated endocytosis in Lowe syndrome fibroblasts. (**A**–**C**) Global lifetime distribution of AP-2 subunit μ2-GFP spots (reflecting endocytic clathrin-coated pits), as assessed by TIRF microscopy in control (**A**), patient cells (**B**) and patient cells rescued with mCherry-OCRL[WT] (**C**) (Ordinate represents the relative frequency spots at each time point). Histograms were fitted with a Gaussian mixture model identifying three subpopulations of endocytic clathrin-coated pits: short- (mean 40 s, blue), medium- (mean 108 s, green) and long-lived (mean 244 s, red). Note the increase in size of the long-lived subpopulation (red) in patient cells (**B**), which is rescued by the re-expression of mCherry-OCRL[WT] (**C**), (n of events = 8657, 9605, and 6930 respectively; n of cells examined = 12, 12 or 7, respectively). (**D**–**F**) Spinning disk analysis of endocytic clathrin-coated pits in control and patient fibroblasts expressing RFP-CLC and

*Figure 4. Continued on next page*

*Figure 4. Continued*

GFP-OCRL$^{WT}$ or GFP-catalytically dead OCRL (GFP-OCRL$^{D523G}$). Representative images are shown in the left panels, kymographs in the middle panels and tracings of the average time courses of fluorescence in the right panels (mean ± SEM, n = 45 events per condition, 3 separate experiments). In control cells (**D**), there is little overlap between red (clathrin) and predominantly green (OCRL) spots, indicating OCRL recruitment when clathrin fluorescence has started to dim. The signal for OCRL then rapidly disappears as the free vesicle moves away from the imaging plane. The same pattern can be observed in patients cells rescued with OCRL$^{WT}$ (**F**), while in patient cells expressing OCRL$^{D523G}$, clathrin lingers for longer times even after the recruitment of OCRL, leading to a substantial overlap of red and green spots (**E**). Scale bar full size: 2 µm; kymograph: 1 min (red: clathrin; green: OCRL). (**G**) Colocalization of clathrin and OCRL fluorescence on individual spots from images such as those shown in the left fields of (**D**–**F**) as determined by a colocalization plugin on ImageJ ('Materials and methods') and reported as Pearson's correlation coefficient (mean ± SEM, n = 7 different cells, 4 separate experiments; two tailed student's t test, ** denotes p = 0.0014 and *** denotes p = 0.0001). (**H**) Representative images (left) and quantification (right) of different stages of clathrin-mediated endocytosis showing a striking increase in the number of U-shaped pits and coated vesicles per µm of cell perimeter (n = 60 cells; two tailed student's t test, ** denotes p = 0.001, *** denotes p = 0.0001 and **** denotes p = 0.00001). (**I** and **J**) A mid-plane confocal image of control (**I**) and patient (**J**) cells showing the internalization of Alexa594-labeled transferrin (Tf) that occurs in 2 min. Note that in patient cells transferrin remains mostly at the cell surface. Scale bar: 10 µm. (**K**) Time course of biotinylated-transferrin internalization in control and patient cells measured by an ELISA-based assay. Uptake is represented as percent of total surface bound biotin-transferrin at 4°C. (n = 3 experiments; two tailed student's t test, ** denotes p = 0.001, and **** denotes p = 0.00001). (**L**) Internal and surface exposed Tf receptor in control and patient cells as revealed by a biotinylation assay, demonstrating the increase of the surface pool of receptor in patient cells.

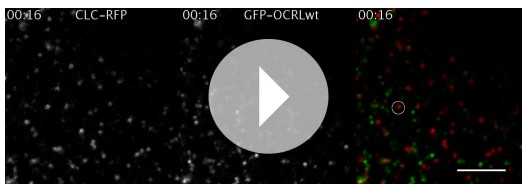

**Video 2**. Clathrin and OCRL dynamics in wild type fibroblasts. Wild-type skin fibroblast co-transfected with GFP-OCRL$^{WT}$ and RFP-CLC showing the recruitment of OCRL to clathrin-coated pits at the end of their lifetime.

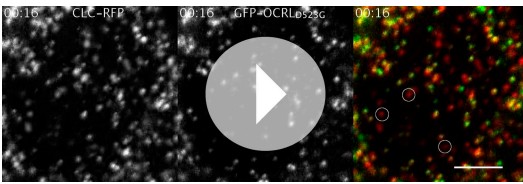

**Video 3**. Clathrin and OCRL dynamics in patient fibroblasts. Lowe syndrome patient fibroblast co-transfected with the catalytically inactive GFP-OCRL$^{D523G}$ and RFP-CLC showing the presence of OCRL at almost every pit and the lack of dynamic turnover of clathrin and OCRL.

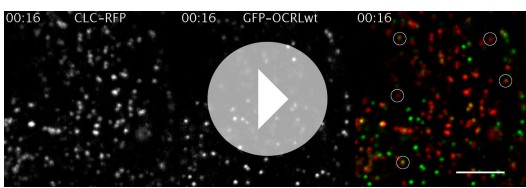

**Video 4**. Expression of wild type OCRL rescues clathrin dynamics in patient fibroblasts. Lowe syndrome patient fibroblast co-transfected with the GFP-OCRL$^{WT}$ and RFP-CLC showing the rescue of clathrin and OCRL dynamics at the pit in the presence of exogenously expressed wild type OCRL.

this possibility, expression of GFP-OCRL$^{WT}$ suppressed the comets (*Figure 5—figure supplement 1G–I'*), while expression of catalytically-inactive GFP-OCRL$^{D523G}$ localized to the head of the comets but failed to rescue this phenotype (*Figure 5C*, *Figure 5—figure supplement 1J–J'*; *Video 6*).

Further analysis of proteins at the head of the comets revealed the presence of late-stage endocytic clathrin coat components and their accessory factors, including clathrin, AP-2, SNX9, SNX18, and PI3KcIIα, as well as transferrin, a known cargo of clathrin-mediated endocytosis (*Figure 5C*; *Video 7*). Nearly every actin tail propelled vesicles positive for these proteins (*Figure 5D*). This was also the case for endogenous proteins, as revealed by the presence of SNX9 and clathrin light chain immunoreactivity at the tips of comets labeled by phalloidin (an F-actin marker) (*Figure 5E*). The tips of these comets however, were mostly devoid of endosomal markers, such as Rab5 or the FYVE domain of HRS (*Figure 5—figure supplement 1K–L*).

The levels of SNX9 were strikingly increased in patient cells (*Figure 5F–G*), possibly due to its accumulation and thus sequestration at comet tips. This may also explain the great increase of SNX9 punctate immunoreactivity in patient cells previously mentioned (*Figure 3I–J*). In contrast, the overall cellular levels of clathrin and AP-2 were similar in control and patient cells (*Figure 5F*). However, an increase in the pool of assembled clathrin coats in patient cells was demonstrated biochemically (*Figure 5H–I*). Homogenization of cells under conditions known to preserve assembled clathrin, followed by high-speed centrifugation to yield cytosol (C) and total particulate (P) fractions, revealed an increased fraction of clathrin heavy chain in the particulate fraction of patient cells (*Figure 5H–I*).

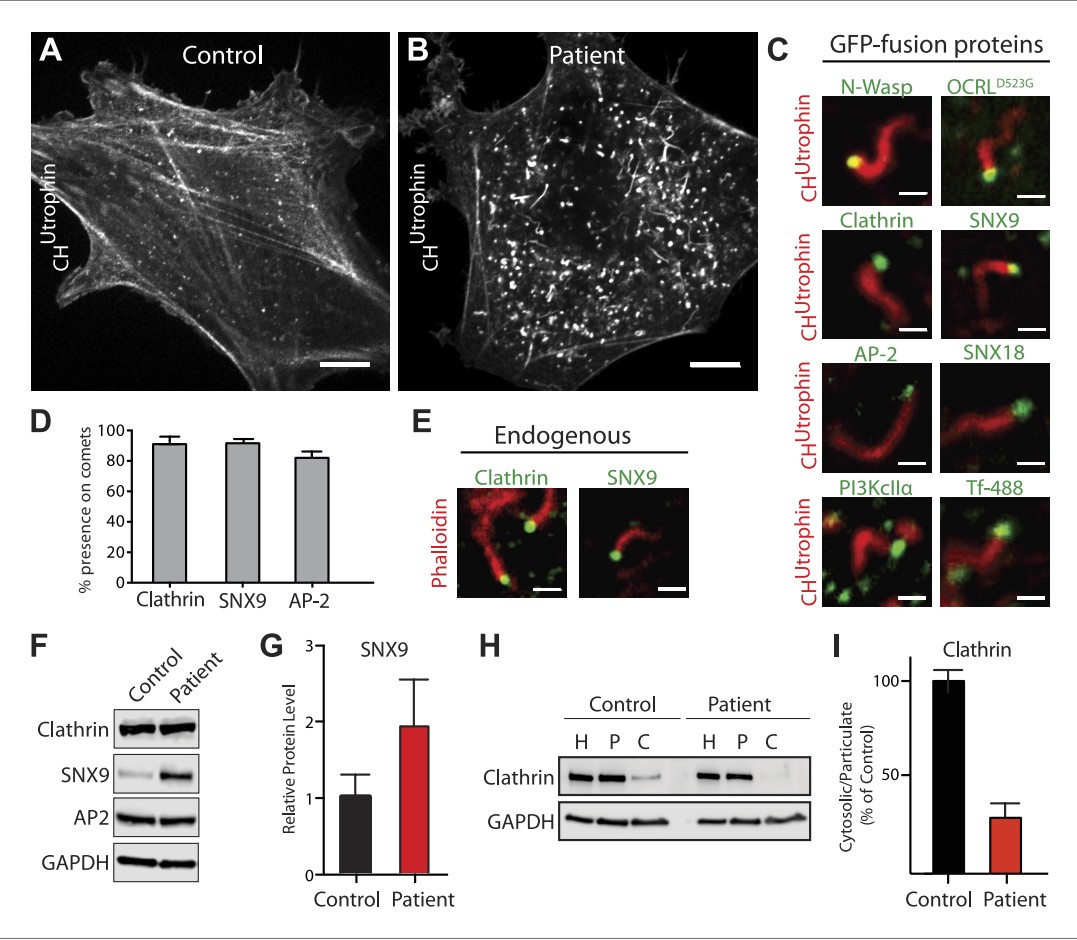

**Figure 5**. Accumulation of endocytic clathrin coat components on intracellular vesicles pushed by actin comets. (**A** and **B**) Fluorescence of mCherry-CH$^{Utrophin}$ (calponin homology domain of Utrophin, an F-actin probe) in control (**A**) and patient (**B**) cells revealing the loss of stress fibers and accumulation of actin tails in Lowe syndrome fibroblasts. Scale bar: 10 µm. (**C**) Spinning disk confocal images of Lowe syndrome fibroblast co-expressing mCherry-CH$^{Utrophin}$ and the proteins indicated, which are all at the tips of the comets. Scale bar: 1 µm. (**D**) Quantification of the percentage of comets positive for the indicated proteins at their tip (n = 500 comets, 5 different cells). (**E**) Immunofluorescence staining of patient cells for phalloidin (an F-actin marker) and clathrin or SNX9, showing the endogenous presence of these proteins the tips of the comets. Scale bar: 1 µm. (**F** and **G**) Western blot analysis of control and patient cells for the indicated proteins showing increased levels of SNX9 in patient cells. Quantification of SNX9 is shown in (**G**) (mean ± SEM, n = 4 experiments). (**H** and **I**) Total particulate [P] and cytosolic [C] fractions derived from control or patient cell homogenates [H] followed by immunoblot analysis (**H**) show a virtual disappearance of the cytosolic pool of unassembled clathrin in patient cells (**I**). ([H]: homogenate; [P]: particulate; [C]: cytosolic).

The following figure supplements are available for figure 5:

**Figure supplement 1**. Perturbation of the actin cytoskeleton in patient cells.

**Figure supplement 2**. SNX9 knockdown in Lowe syndrome fibroblasts.

Thus, the coat and cargo of the intracellular actin-propelled vesicles are very similar to those of endocytic clathrin-coated pits. This supports the idea that vesicles at comet tips are endocytic clathrin-coated vesicles whose coat failed to shed due to the lack of OCRL catalytic activity. A small subset of vesicles driving the comets were larger than would be expected from endocytic clathrin-coated vesicles and may represent micropinosomes.

Given the striking increase of SNX9 levels in patient cells and its known role in coordinating clathrin-coated pits maturation with actin nucleation (*Yarar et al., 2007*), we explored its importance in the

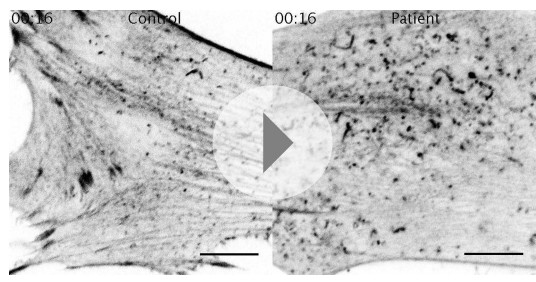

**Video 5**. Comets in Lowe syndrome patient fibroblasts. Control and Lowe syndrome patient fibroblast transfected with the F-actin probe mCherry-CH[Utrophin]. The control cell shows the characteristic stress fibers and small actin puncta typically associated with sites of clathrin-mediated endocytosis. Stress fibers are lost in patient cells, which instead exhibit actin comets that rocket through the cytosol.

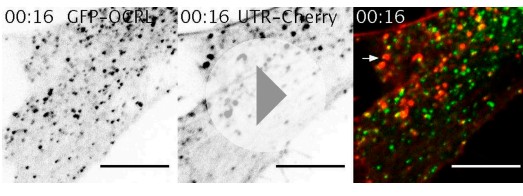

**Video 6**. Catalytically inactive OCRL localizes to the tips of actin comets. Lowe syndrome patient fibroblast co-transfected with the F-actin probe mCherry-CH[Utrophin] and the catalytic dead GFP-OCRL[D523G] showing that the machinery necessary to recruit OCRL is present on the vesicles at the tip of the comets.

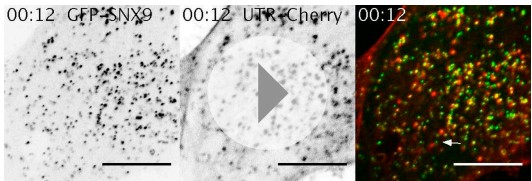

**Video 7**. SNX9 is found on vesicles pushed by actin comets. Lowe syndrome patient fibroblast co-transfected with the F-actin probe mCherry-CH[Utrophin] and GFP-SNX9 revealing the presence of this endocytic factor on the vesicles at the tip of the comets.

formation of actin tails. Knockdown of SNX9 followed by immunostaining with phalloidin revealed a decrease in the number of endogenous comets observed in patient cells (***Figure 5—figure supplement 2A–C***). There was noticeable cell-to-cell variability however, roughly correlating with residual SNX9 levels in knockdown cells (generally higher in patient cells given the elevated basal level of this protein). Importantly, most of the remaining SNX9 puncta observed in cells after knockdown associated with the tips of the comets (***Figure 5—figure supplement 2C***). These results suggest that the interaction between SNX9 and OCRL may help coordinate actin polymerization at late stage clathrin-coated pits with the turnover of PI(4,5)$P_2$. The recruitment of OCRL to clathrin-coated pits was not dependent on its interaction with SNX9, however. This was evidenced by the expression of an SNX9-binding mutant construct of OCRL (GFP-OCRL[P3]) that still localized to clathrin-coated pits in patient cells (***Figure 5—figure supplement 2D***). Given the interactions of OCRL with several prominent endocytic factors such as clathrin and AP-2, it is highly likely that multiple binding partners regulate its recruitment to clathrin-coated pits.

## Progressive loss of actin comets upon plasma membrane PI(4,5)$P_2$ depletion

If actin comets are in fact nucleated by plasma membrane-derived PI(4,5)$P_2$ still present on un-coated endocytic vesicles due to the absence of OCRL, depletion of plasma membrane PI(4,5)$P_2$ should result in their progressive disappearance. Alternatively, if comets are generated by the synthesis of PI(4,5)$P_2$ on intracellular structures, depletion of plasma membrane PI(4,5)$P_2$ would not affect them. In order to discern between these two scenarios, we selectively depleted PI(4,5)$P_2$ at the plasma membrane via the activation of a PLC-coupled receptor, the $M_1$ muscarinic acetylcholine receptor ($M_1$R).

Patient cells co-expressing untagged $M_1$R, the PI(4,5)$P_2$ probe GFP-PH[PLCδ] and the F-actin reporter mCherry-CH[UTR] were analyzed by dual-color time-lapse confocal microscopy while an agonist (OxoM), and subsequently an antagonist (atropine), were added during live acquisition. Prior to OxoM addition, GFP-PH[PLCδ] was selectively localized at the plasma membrane, as typically observed for this PI(4,5)$P_2$ biosensor in resting cells (***Figure 6A***). The addition of OxoM triggered PLC-mediated PI(4,5)$P_2$ hydrolysis into IP3 and diacyl glycerol, which resulted in the relocation of GFP-PH[PLCδ] to the cytosol, indicating PI(4,5)$P_2$ depletion at the plasma membrane (***Figure 6A***). This change correlated with a transient increase of cytosolic $Ca^{2+}$, as expected (***Figure 6C***; ***Horowitz et al., 2005***). In turn, this led to a massive transient depolymerization of actin, including actin comets, and a sharp increase in the diffuse cytosolic fluorescence of the actin reporter, mCherry-CH[UTR] (***Figure 6B–D***). Subsequently, as $Ca^{2+}$ levels reverted to the baseline level (***Figure 6C***), actin comets reformed (***Figure 6B–D***), often reemerging at sites where

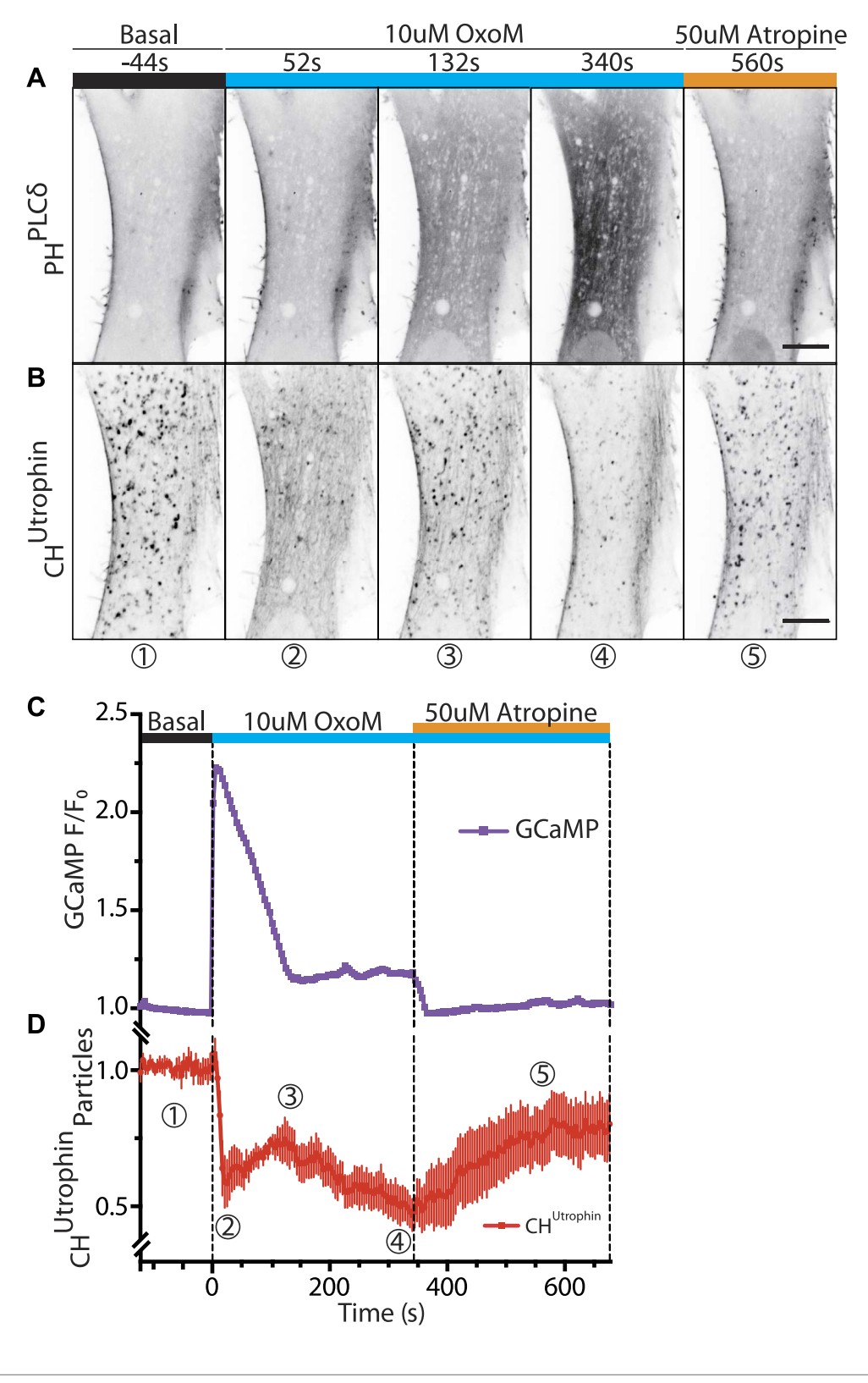

**Figure 6**. Vesicles propelled by actin tails originate from the plasma membrane. Patient fibroblasts co-expressing the PI(4,5)P$_2$ probe GFP-PH$^{PLC\delta}$, mCherry-CH$^{Utrophin}$ and untagged muscarinic (M$_1$R) receptor. (**A** and **B**) Sequential confocal images of GFP-PH$^{PLC\delta}$ and mCherry-CH$^{Utrophin}$ fluorescence (fluorescence signal is reversed to appear dark). 10 μM

*Figure 6. Continued on next page*

*Figure 6. Continued*

OxoM was added at time zero and 50 µM atropine (an $M_1R$ antagonist) 344 s later. The PI(4,5)P$_2$ reporter (GFP-PH$^{PLC\delta}$) shifts reversibly from the cell periphery to the cytosol in response to OxoM. mCherry-CH$^{Utrophin}$ foci represent actin comets that transiently disassemble during the Ca$^{2+}$ peak, reassemble, and then gradually disappear as PI(4,5)P$_2$ is depleted leading to impaired endocytosis. Scale bar: 10 µm. (**C**) Fluorescence of GCaMP3, a Ca$^{2+}$ sensor, showing a transient spike in cytosolic Ca$^{2+}$ upon plasma membrane PI(4,5)P$_2$ hydrolysis in response to the $M_1R$ agonist OxoM, which triggers PLC activity. (**D**) Quantification of the number of comets from thresholded mCherry-CH$^{Utrophin}$ fluorescence during the application of OxoM and atropine to patient cells (mean ± SEM, n = 5 experiments). Encircled numbers correspond to the different five stages represented in (**B**).

The following figure supplement is available for figure 6:

**Figure supplement 1**. Comets reemerge from the same sites following transient actin depolymerization.

they had previously disappeared (*Figure 6—figure supplement 1*). This suggested that vesicles driving the comets had retained actin-nucleating properties during the transient actin depolymerization.

In the continued presence of OxoM (conditions where plasma membrane PI(4,5)P$_2$ remains depleted), reappearance of the comets after the Ca$^{2+}$-induced actin depolymerization was followed by their progressive loss (*Figure 6B,D*; *Video 8*). Termination of the OxoM signal by addition of its antagonist, atropine, led to the rapid reformation of PI(4,5)P$_2$ at the plasma membrane (*Figure 6A*) and to a striking progressive recovery of the number of new comets rocketing throughout the cytosol (*Figure 6B,D*; *Video 8*). These results are consistent with the idea that the bulk of the vesicles found at the tip of actin comets are newly formed endocytic vesicles originating from the plasma membrane.

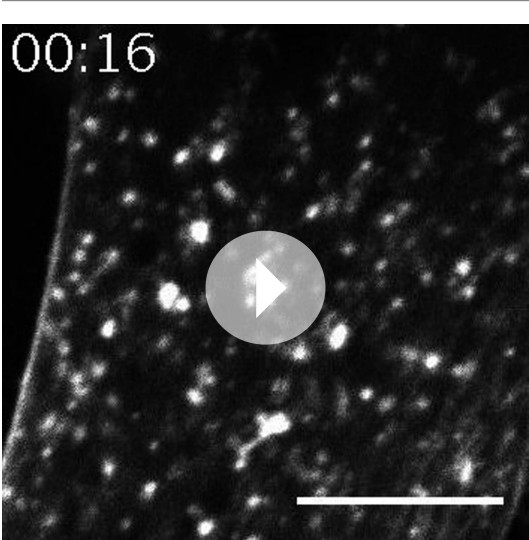

**Video 8**. Selective depletion of plasma membrane PI(4,5)P$_2$ abolishes the comets. Lowe syndrome patient fibroblast co-transfected with the F-actin probe mCherry-CH$^{Utrophin}$, GFP-PH$^{PLC\delta}$ (not shown) and the untagged muscarinic receptor $M_1R$. The cell is treated with 10 µM OxoM at 1m44s, which stays in the imaging media for the duration of the acquisition, followed by 50 µM atropine at 7m24s. Upon the addition of OxoM there's a rapid depolymerization of F-actin due to a rise in cytosolic calcium followed by the rapid reformation of the comets. Upon the continued presence of OxoM however, the comets gradually disappear due to the continued depletion of plasma membrane PI(4,5)P$_2$. This effect is reversed upon the addition of the antagonist Atropine, which allows the reformation of PI(4,5)P$_2$ at the plasma membrane, leading to the resurgence of the actin comets.

## PI(4,5)P$_2$ effectors compete with PH$^{PLC\delta}$ at the tip of actin comets

The observations described thus far suggest a scenario in which the lack of OCRL-mediated PI(4,5)P$_2$ hydrolysis results in a delay of endocytosis and impaired uncoating of endocytic clathrin-coated vesicles. They further suggest that coated PI(4,5)P$_2$-rich vesicles drive actin nucleation. This hypothesis, however, contrasts with the very infrequent detection of the PI(4,5)P$_2$ probe, GFP-PH$^{PLC\delta}$, at the tip of comets (*Figure 7A*, inset). This was particularly evident in images through the middle of the cell revealing the robust presence of PI(4,5)P$_2$ at the plasma membrane but not intracellularly (*Figure 7A*). Even a higher sensitivity probe, consisting of two tandem PH domains of PLCδ (GFP-2XPH$^{PLC\delta}$) yielded similar results. The difference in GFP-PH$^{PLC\delta}$ labeling of comet tips and of the plasma membrane was even more striking on images of the bottom plane of a cell showing intracellular comets that encountered the plasma membrane and evaginated it into filopodia-like structures, akin to those generated by *Listeria*-driven comets (*Figure 7B,D*; *Video 9*). In this case, the heavy labeling of the plasma membrane of filopodia encapsulating the actin comet stood in striking contrast to the lack of labeling (or very weak labeling) of the vesicle at the tip of the comet.

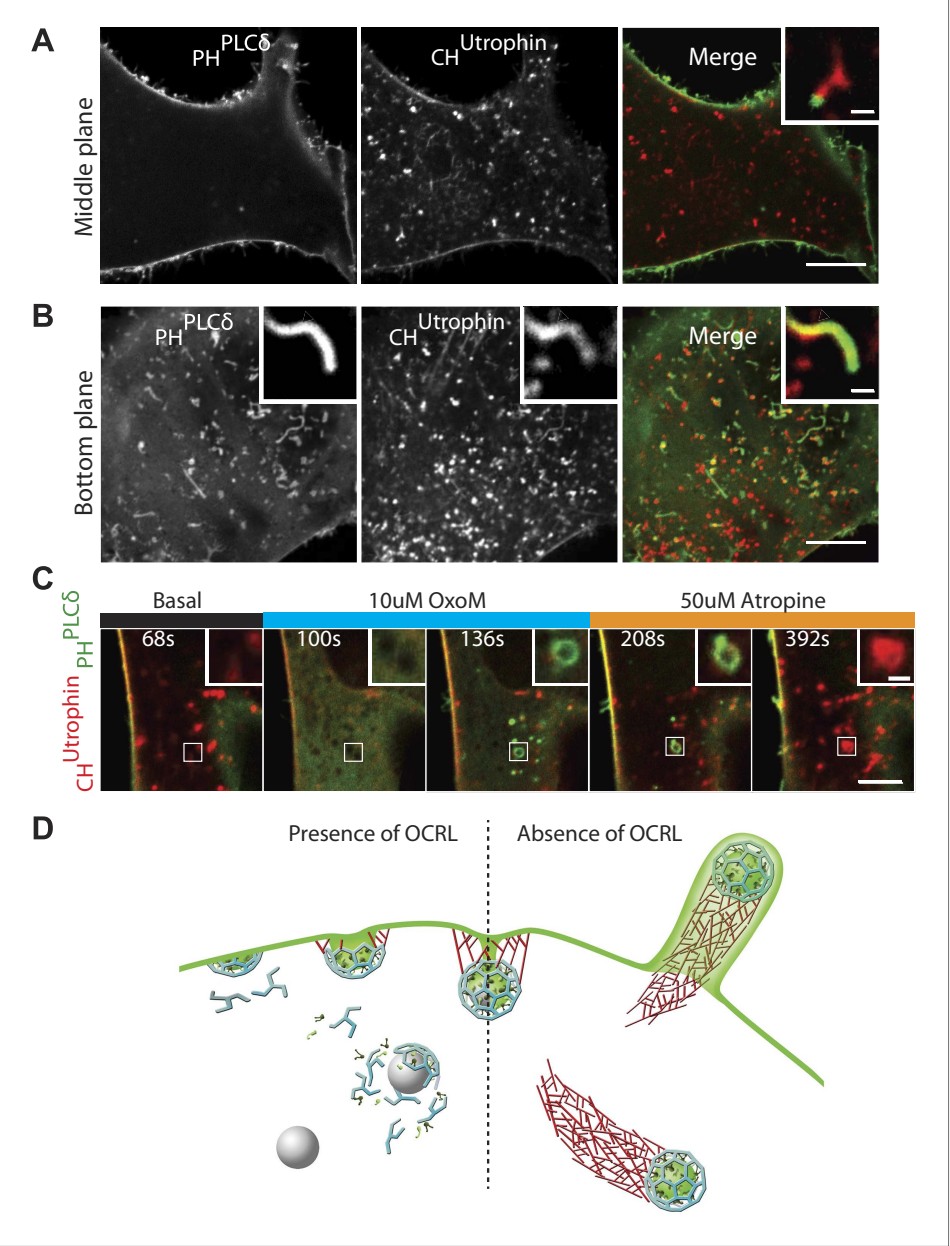

**Figure 7**. PI(4,5)P$_2$ detection on intracellular vesicles. (**A–B**) Confocal images of patient fibroblasts co-expressing GFP-PH$^{PLC\delta}$ and mCherry-CH$^{Utrophin}$ at two different focal planes, midsection and bottom surface, are shown in (**A**) and (**B**) respectively. The midsection image (**A**) shows robust presence of the PI(4,5)P$_2$ probe at the plasma membrane but only occasional presence at the tip of the comets. One example is shown in the inset. The bottom surface (**B**) image shows numerous PI(4,5)P$_2$ positive cell evaginations resulting from comets pushing into the plasma membrane. One example is shown in the inset. As the comet pushes into the plasma membrane it becomes enveloped in PH$^{PLC\delta}$-positive plasma membrane. Scale bar full size: 10 µm; inset: 1 µm. (**C**) Sequential confocal images of a patient cell expressing untagged M$_1$R, GFP-PH$^{PLC\delta}$ and mCherry-CH$^{Utrophin}$ showing that in response to plasma membrane PI(4,5)P$_2$ hydrolysis and increase in cytosolic Ca$^{2+}$ induced by OxoM, actin comet disassemble and the PI(4,5)P$_2$ probe relocates to intracellular vesicles. As actin reassembles, mCherry-CH$^{Utrophin}$ displaces the PI(4,5)P$_2$ probe. (**D**) Model depicting the impact of the lack of OCRL in Lowe syndrome patients on clathrin-mediated endocytosis. Absence of the 5-phosphatase activity of OCRL delays endocytosis by stalling late-stage pits, impairs uncoating and triggers the nucleation of comets from clathrin vesicles that retain PI(4,5)P$_2$ and fail to uncoat.

The following figure supplement is available for figure 7:

**Figure supplement 1**. Overexpression of PIP5K1γ (a PI4P 5-kinase) in wild type fibroblasts induces actin comets.

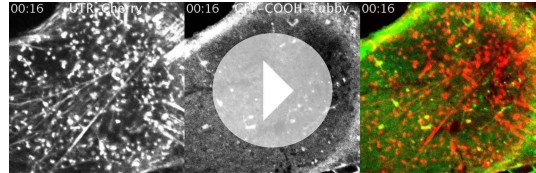

**Video 9**. Actin comets come into contact with and evaginate the plasma membrane. Lowe syndrome patient fibroblast co-transfected with the F-actin probe mCherry-CHUtrophin and GFP-PHPLCδ showing the presence of comets that come into contact with the plasma membrane and evaginate it to create filopodia-like structures, akin to those observed upon Lysteria infection. Note that comets are not positive for GFP-PHPLCδ until they come into contact with the GFP-PHPLCδ positive plasma membrane. At that point the comet pushes out the plasma membrane and is enveloped by it (see cartoon of *Figure 7*).

Since these results brought into question the presence of $PI(4,5)P_2$ at the tips of comets, we examined the localization of the $PI(4,5)P_2$ probe in an additional cellular model of $PI(4,5)P_2$-dependent actin comet formation, the overexpression of a PI4P 5-kinase. Overexpression of one such enzyme, PIPkinase type-1-gamma (PIPK1γ), along with GFP-PH$^{PLCδ}$ and mCherry-CH$^{UTR}$ in control human fibroblasts cells led to a robust formation of actin comets (*Figure 7—figure supplement 1A–A"*). Even in this case, where the intracellular pool of $PI(4,5)P_2$ was dramatically expanded, detection of GFP-PH$^{PLCδ}$ labeling at the tips of the comets (*Figure 7—figure supplement 1B*) was inconsistent (although more frequently observed than in Lowe syndrome patient cells), in contrast to the robust labeling of the plasma membrane (*Figure 7—figure supplement 1C–C"'*).

This finding could have three non-mutually exclusive explanations: (1) $PI(4,5)P_2$ may be present at a much lower concentration on the vesicles than on the plasma membrane, but these low levels may be sufficient to nucleate actin, (2) GFP-PH$^{PLCδ}$ may be a better reporter of plasma membrane $PI(4,5)P_2$ given the presence of a yet unidentified PH$^{PLCδ}$ co-receptor only on this membrane; (3) $PI(4,5)P_2$ present on the vesicles may be shielded and possibly sequestered by actin and actin regulatory proteins.

One observation favoring the idea that actin and actin regulatory proteins may shield/sequester $PI(4,5)P_2$ came from patient cells overexpressing $M_1R$ and treated with OxoM. In such cells, several vesicles became positive for GFP-PH$^{PLCδ}$ during the period of actin depolymerization that accompanied the transient rise in cytosolic $Ca^{2+}$ (*Figure 7C*). However, GFP-PH$^{PLCδ}$ was rapidly displaced by mCherry-CH$^{UTR}$ as actin comets reformed, providing further evidence for the presence of $PI(4,5)P_2$ on internal vesicles in Lowe syndrome patient cells (*Figure 7C*).

Overall, these findings support a model in which OCRL acts at late-stage clathrin-coated pits to dephosphorylate $PI(4,5)P_2$ and facilitates the shedding of endocytic factors (*Figure 7D*). In the absence of OCRL, as in Lowe syndrome, the accumulation of $PI(4,5)P_2$-rich vesicles that fail to lose their clathrin coat leads to an endocytic defect resulting from the sequestration of endocytic factors on these vesicles (*Figure 7D*).

## Discussion

Enzymes containing a 5-phosphatase domain have been implicated at multiple stages of clathrin-mediated endocytosis, from clathrin coat nucleation to uncoating (*Pirruccello and De Camilli, 2012*). While OCRL, which directly interacts with clathrin, had been detected at late stage clathrin-coated pits (*Erdmann et al., 2007*; *Mao et al., 2009*), so far there has been no evidence from loss-of-function studies for a direct impact of this enzyme on the dynamics of endocytic clathrin-mediated budding. Data concerning the action of OCRL in membrane traffic, and the impact of its loss in patient cells, pointed primarily to a post-endocytic role, including intracellular clathrin-dependent transport reactions (*Dressman et al., 2000*; *Ungewickell et al., 2004*; *Choudhury et al., 2005*; *Hyvola et al., 2006*; *Swan et al., 2010*; *Vicinanza et al., 2011*). Additional roles for OCRL in clathrin-independent forms of endocytosis (micropinocytosis and phagocytosis) had also been reported (*Bohdanowicz et al., 2012*; *Kuhbacher et al., 2012*; *Marion et al., 2012*). Our present results demonstrate that the absence of OCRL in Lowe syndrome patient cells results in an impairment of clathrin-mediated endocytosis. These findings provide new insights into the basic mechanisms of endocytosis as well as into the molecular pathogenesis of Lowe syndrome and Dent's disease.

A role of $PI(4,5)P_2$ dephosphorylation in the normal progression of clathrin-mediated endocytosis was first described at neuronal synapses, where studies of synaptojanin 1 emphasized an uncoating function of this protein (*Cremona et al., 1999*). Subsequent work in a variety of model systems supported this idea but also suggested a role of this phosphoinositide in the late stages of endocytosis.

Dephosphorylation of PI(4,5)$P_2$ in the membranes of the budding vesicles was proposed to act as a mechanism to generate a line tension at the vesicle neck (*Stefan et al., 2002*; *Rusk et al., 2003*; *Mani et al., 2007*; *Sun et al., 2007*; *Liu et al., 2009*; *Chang-Ileto et al., 2011*). Progression through the late stages of endocytosis could be also affected by the sequestration of endocytic proteins on PI(4,5)$P_2$-rich uncoated vesicles, as postulated for synapses lacking synaptojanin 1 or all three endophilins (*Mani et al., 2007*; *Milosevic et al., 2011*).

The present study demonstrates a defect of endocytosis in Lowe syndrome patient fibroblasts. This defect is characterized by an accumulation of late-stage (primarily U-shaped) endocytic clathrin-coated pits and by a delay in the internalization of transferrin, a clathrin-coated pit cargo. Our results also point to a defect in clathrin uncoating, as in the case of synaptojanin 1 KO synapses. More specifically, electron microscopy demonstrated a robust increase in the number of clathrin-coated vesicles in patient cells relative to control fibroblasts. Additionally, immunofluorescence and live microscopy analysis revealed an abundance of intracellular vesicles positive for both coat and cargo proteins of endocytic origin including, besides clathrin, the adaptor complex AP-2, SNX9, PI3KcIIα and transferrin. These vesicles were at the tips of actin comets, which are known to be nucleated by PI(4,5)$P_2$-positive organelles. The dramatic decrease in the number of these comets upon the selective, pharmacologically-induced depletion of PI(4,5)$P_2$ at the plasma membrane demonstrates that at least a large cohort of these comets are nucleated by endocytic vesicles. The prominent impact of the lack of OCRL on total cellular PI(4,5)$P_2$ levels in patient cells is reminiscent of the robust increase in brain levels of this phosphoinositide observed in the absence of synaptojanin 1 (*Cremona et al., 1999*). We suggest that synaptojanin 1 and OCRL have partially overlapping functions, but with predominant roles at synapses and in non-neuronal cells, respectively.

The newly discovered direct interaction of OCRL with SNX9, a protein implicated in the transition from late stage clathrin-coated pits to free clathrin-coated vesicles, is of significant interest as it provides a new link between OCRL function and late-stage endocytic clathrin-coated pit nucleation. Interestingly, SNX9 also binds synaptojanin 1 and 2, consistent with a partial overlap in the function of these 5-phosphatases (*Yeow-Fong et al., 2005*). SNX9 is a member of a family of BAR domain-containing endocytic adaptors that help coordinate changes in bilayer curvature with reactions that occur as membranes progress along the endocytic pathway. Specifically, SNX9 is thought to have an important role in coupling clathrin coat maturation to actin nucleation (*Yarar et al., 2007*). Our observation that the knockdown of SNX9 decreases comet formation in patient cells supports a role for this protein in actin nucleation at clathrin-coated pits. It also suggests a functional link between SNX9-regulated actin polymerization and the turnover of PI(4,5)$P_2$ controlled by OCRL. The modular structure of SNX9 is ideally suited to mediate this coordination. SNX9 binds the plasma membrane with a PX-BAR domain module, which exhibits a preference for curved membranes enriched in PI(4,5)P2 and 3-phosphorylated phosphoinositides (*Gallop et al., 2013*), clathrin via a 'clathrin-box' motif and AP-2 via an α-adaptin-binding motif (*Lundmark and Carlsson, 2003*). Through its SH3 domain, SNX9 binds and thus helps to activate both N-WASP to trigger actin nucleation, and dynamin to mediate fission (*Yarar et al., 2007*). The SH3 domain of SNX9 binds synaptojanin and also OCRL, as we have demonstrated in this study. Given that SNX9 is a protein dimer that is present in multiple copies at endocytic sites, SNX9 is an attractive candidate to help bring together all these factors.

Recently, SNX9 was shown to act as an effector of PI(3,4)$P_2$, which is generated at endocytic clathrin-coated pits by PI3KcIIα (*Posor et al., 2013*). In principle, 5-phosphatases such as OCRL and synaptojanin could directly contribute to PI(3,4)$P_2$ production by dephosphorylation of PI(3,4,5)$P_3$, or indirectly by generating PI4P, the PI3KcIIα substrate, from PI(4,5)$P_2$ dephosphorylation (*Posor et al., 2013*; *Schmid and Mettlen, 2013*). These models would place OCRL upstream of SNX9. However, SNX9 is recruited to coated pits prior to OCRL, and also in its absence. Instead, the recruitment of OCRL at endocytic clathrin-coated pits coincides with the temporal recruitment signature of GAK (*Taylor et al., 2011*), which, like its homologue auxilin, is a cofactor for Hsc70 and thus a key player in the disassembly of the clathrin lattice after fission (*Fotin et al., 2004*; *Massol et al., 2006*). Both OCRL and GAK peak after the clathrin fluorescence has started to diminish, as expected for proteins that mediate the shedding of the clathrin coat (GAK) and its adaptors (OCRL). Interestingly, the lack of GAK or auxilin results in endocytic defects in addition to the accumulation of clathrin-coated vesicles and empty clathrin cages (*Lee et al., 2005*, *Yim et al., 2005*). This impairment of endocytosis is thought to result from the sequestration of coat components on assembled lattices. Similarly, a defect in uncoating in the absence of OCRL may impair the recycling of endocytic factors and affect the maturation of

clathrin-coated pits. As OCRL binds multiple proteins present in endocytic clathrin coats, an important question for future studies will be to determine the precise mechanisms responsible for the timing of OCRL recruitment.

The presence in Lowe syndrome fibroblasts of numerous actin comets driven by clathrin-positive vesicles adds further support to a close relationship between endocytic clathrin coats and actin, a topic of intense debate, although most recent studies support such a link at least for a major subset of clathrin-dependent endocytic events (*Fujimoto et al., 2000*; *Merrifield et al., 2002, 2005*; *Boucrot et al., 2006*; *Kaksonen et al., 2006*; *Tsujita et al., 2006*; *Ferguson et al., 2009*; *Boulant et al., 2011*). Actin has been implicated in both the invagination of clathrin-coated pits as well as in the propulsion of endocytic vesicles away from endocytic sites. The occurrence of comets in Lowe syndrome cells suggests that failure to dephosphorylate $PI(4,5)P_2$ not only delays coat shedding, but also disrupts the release of actin nucleating factors associated with endocytic clathrin coats.

The collective information emerging from this and previous studies suggests that the function of OCRL is needed, either directly or indirectly, at a multiplicity of sites. The 5-phosphatase activity of OCRL is required to complete endocytosis and to prevent the ectopic accumulation of $PI(4,5)P_2$ on endosomes and membranes in the Golgi complex area (this work; *Choudhury et al., 2005*; *Dressman et al., 2000*; *Erdmann et al., 2007*; *Hyvola et al., 2006*; *Swan et al., 2010*; *Ungewickell et al., 2004*; *Vicinanza et al., 2011*). This multiplicity of actions is supported by its numerous interactors, which comprise proteins implicated in membrane traffic, signaling and actin regulation in a variety of cellular compartments (*Pirruccello and De Camilli, 2012*). As shown here, proteins implicated in clathrin-mediated endocytosis are prominently featured in the OCRL 'interactome', which also comprises numerous proteins that participate in the clathrin-dependent traffic between endosomes and the Golgi complex. These include the cation-independent mannose 6-phosphate receptor (CI-M6PR) and the clathrin adaptor EpsinR, implicating OCRL in these trafficking stations as well. It remains unclear whether the function of OCRL at these intracellular sites is to prevent the buildup of $PI(4,5)P_2$ or to control small yet physiologically relevant $PI(4,5)P_2$ pools.

A surprising result of our study was the unexpected absence of a robust signal for the $PI(4,5)P_2$ probe, PH[PLCδ], at intracellular sites in Lowe syndrome fibroblasts. The presence of clathrin, AP-2 and actin nucleating proteins on intracellular vesicles, however, strongly suggests the presence of $PI(4,5)P_2$ on these vesicles. Sequestration of $PI(4,5)P_2$ on such vesicles by clathrin adaptors and actin nucleating factors may be one reason explaining absent or inefficient PH[PLCδ] labeling, as we have shown that transient depolymerization of actin results in the appearance of the $PI(4,5)P_2$ reporter on intracellular vesicles.

OCRL has a close homologue, INPP5B, and these two proteins share many common interactors including endosomal proteins and Rabs (*Shin et al., 2005*; *Hyvola et al., 2006*; *Erdmann et al., 2007*; *Fukuda et al., 2008*). INPP5B however, does not contain either clathrin or AP-2 binding sites (*Ungewickell et al., 2004*; *Choudhury et al., 2005*). Correspondingly, OCRL and INPP5B share similar localizations on endosomes and in the Golgi complex area but INPP5B is not present at endocytic clathrin-coated pits (*Erdmann et al., 2007*; *Williams et al., 2007*; *Mao et al., 2009*). Thus, a major functional difference between OCRL and INPP5B seems to be the involvement of OCRL, but not of INPP5B, in clathrin-mediated endocytosis. One important open question is why the lack of OCRL produces major phenotypes in humans but not in mice, in spite of the conservation of all known properties of OCRL in mice and humans, including the SNX9 binding site reported here. As INPP5B does not participate in endocytic clathrin-coated dynamics, differences in the expression or splicing of INPP5B in mice and humans cannot fully account for this difference. Elucidating mechanisms underlying this difference represents an important goal of future studies.

Previous studies had attributed the proximal tubule reabsorption defects occurring in Lowe syndrome and Dent's disease primarily to abnormal endosomal recycling, such as for example the recycling of megalin, the scavenger receptor that mediates low molecular weight protein endocytosis, and CI-M6PR (*Erdmann et al., 2007*; *Vicinanza et al., 2011*; *Mehta et al., 2014*). Our findings suggest that endocytosis itself may also be affected, as shown previously for CLC5 mutations that produce Dent's disease (*Novarino et al., 2010*) although they do not exclude the previously suggested intracellular sorting abnormalities (*Lowe, 2005*; *Vicinanza et al., 2011*). Rather, they suggest that defects in endocytosis and in the fate of newly formed endocytic vesicles ultimately contribute to abnormal cargo sorting, as exemplified by the accumulation of clathrin-coated vesicles at the tip of actin comets. Impairments in endocytosis may also contribute to the cognitive deficiencies observed in Lowe

syndrome patients. While synaptojanin 1 is the key player in clathrin-mediated endocytosis at synapses, OCRL may function at clathrin-coated pits that participate in a housekeeping form of endocytosis in neurons.

## Materials and methods

### Plasmids and reagents

GFP-Rab5, mRFP-clathrin light chain (mRFP-CLC), GFP-N-Wasp, GFP-OCRL, catalytically inactive GFP-OCRL[D523G], deletion fragments GST-OCRL[176] and GST-OCRL[217] and clathrin-binding mutant GFP-OCRL[X3] have been described previously (*Perera et al., 2006*; *Erdmann et al., 2007*; *Ferguson et al., 2009*; *Mao et al., 2009*; *Idevall-Hagren et al., 2012*). mCherry-tagged OCRL was generated by excising the OCRL coding sequence from GFP-OCRL and ligating it into the pmCherry-C1 vector (Clontech Laboratories, Inc., Mountain View, CA) using XhoI and BamHI. Three prolines in OCRL (P177, P178 and P181) were mutated to alanine using site-directed mutagenesis (QuickChange II-XL; Agilent Technology, Santa Clara, CA) and GFP-OCRL as the original template to generate GFP-OCRL[P3]. The following primers were used: OCRL-P177,78,81A_S 5'-GGGATTCATCGGGAAGCCGCACCTCCAGCCTTTTCAGT-3' and OCRL-P177,78,81A_AS 5'-ACTGAAAAGGCTGGAGGTGCGGCTTCCCGATGAATCCC-3'.

The following plasmids were kind gifts: GFP-SNX9 (Kai Erdmann, University of Sheffield, United Kingdom), mCherry-CH[Utrophin] (William Bement, University of Wisconsin, Madison, WI), GFP-PH[PLCδ] (Antonella De Matteis, Telethon Institute of Genetics and Medicine, Naples, Italy), µ2-GFP (Alexander Sorkin, University of Pittsburgh, Pittsburgh, PA), GFP-PI3KcIIα (Jan Domin, University of Bedfordshire, Luton, UK), GFP-2X-FYVE[HRS] (Harald Stenmark, University of Oslo, Oslo, Norway), $M_1$ muscarinic acetylcholine receptor ($M_1$R) (Bertil Hille, University of Washington, Seattle, WA), and pCAG-GCaMP5 (Loren Looger, Howard Hughes Medical Institute, Janelia Farm Research Campus, Ashburn, VA).

Transferrin-Alexa 488 and 594 as well as biotinylated transferrin were purchased from Molecular Probes (Life Technologies, Carlsbad, CA). Avidin, biocytin, OxoM and Atropine were obtained from Sigma (Sigma-Aldrich, St. Louis, MO) and the Arp2/3 inhibitor CK-666 was purchased from Calbiochem (EMD Millipore Corporation, Billerica, MA).

### Antibodies

Antibodies for immunofluorescence and immunoblotting were obtained from commercial sources: rabbit anti-OCRL, rabbit anti-PI3KC2A, rabbit anti-CI-M6PR/IGF2R, mouse anti-tubulin, mouse anti-beta actin, (Sigma-Aldrich, St. Louis, MO), rabbit anti-clathrin light chain 4045 (EMD Millipore Corporation, Billerica, MA), mouse anti-clathrin heavy chain TD1 (American Type Culture Collection, Manassas, VA), mouse anti-alpha adaptin AP6 (Affinity BioReagents, Golden, CO), rabbit anti-EpsinR/CLINT1 (Bethyl Laboratories, Montgomery, TX), mouse anti-GAPDH (Santa Cruz Biotechnology, Dallas, TX), rabbit anti-GFP (Abcam, Cambridge, MA), rabbit anti-p34 (EMD Millipore Corporation, Billerica, MA), mouse anti-Transferrin receptor, goat anti-rabbit IgG Alexa Fluor 488, goat anti-mouse IgG Alexa Fluor 594 and Alex Fluor 594-phalloidin (Life Technologies, Calsbad, CA). Rabbit anti-SNX9 for immunoblotting was generated in our lab, while rabbit anti-SNX9 for immunofluorescence were kind gifts from Sven Carlsson (Umeå University, Umeå, Sweden) and Sandra Schmid (UT Southwestern, Dallas, TX).

### Cell culture, transfection and siRNA

De-identified human dermal fibroblasts derived from punch biopsies as well as HeLa M and HEK293 cells were cultured at 37°C in 10% CO2 in DMEM supplemented with 10% FBS and 100 mg/ml penicillin/streptomycin (Life Technologies, Carlsbad, CA). For imaging, HeLa M and human wild type or patient fibroblasts were electroporated with Amaxa Nucleofector method using solution R/program A-24 (Lonza, Basel, Switzerland). Cells were plated at subconfluent densities into 35 mm glass bottom (thickness = 0.17 mm) dishes (Mattek, Ashland, MA) and cultured for 12 to 24 hr prior to imaging.

Control or patient cells seeded at day 0 were transfected with siRNAs using RNAiMax (Life Technologies) according to the manufacturer's instructions, expanded on day 1, and used for the experiment on day 3. All siRNAs sequences were used as a 27-mers DsiRNAs (IDT, Coralville, IA). The SNX9 sequence used was: 5'-*AACAGTCGTGCTAGTTCCTCATCCA-3'*. For silencing SNX18, the sequence used was: 5'-*GCACCGACGAGAAAGCCTGGAAGCA-3'*. The negative control was a previously validated sequence provided by IDT DNA: 5'-CGTTAATCGCGTATAATACGCGTAT-3'.

## Generation of cell lines for proteomic analysis

HeLa M cells were transfected with either GFP-OCRL or pEGFP-C1 (Clontech Laboratories, Inc., Mountain View, CA) using Lipofectamine 2000 (Life Technologies, Carlsbad, CA). 24 hr post-transfection, cells were plated into selection medium containing 750 µg/ml G418 Sulphate (Gibco BRL, Life Technologies, Carlsbad, CA). Following antibiotic selection, high and low GFP-expressing cells were isolated by fluorescence-activated cell sorting (FACS) and plated at low density to facilitate clonal colony formation. Individual colonies were picked, passaged directly into glass-bottom 35 mm dishes (Mattek Corporation, Ashland, MA) and imaged by SDC microscopy 15–24 hr later to assess the expression level, localization and dynamics of GFP-OCRL. Selected colonies were cultured and samples were then subjected to SDS-PAGE followed by Odyssey quantitative western blotting (LI-COR Biotechnology, Lincoln, NE). Rabbit anti-OCRL antibody (Sigma-Aldrich, St. Louis, MO) was used to assess the ratio of exogenous GFP-OCRL to endogenous OCRL, which was then normalized to tubulin levels using mouse anti-tubulin antibody (Sigma-Aldrich, St. Louis, MO).

## Electron microscopy

Cells were fixed in 2% glutaraldehyde-0.1 M sodium cacodylate. They were post-fixed with 1% $OsO_4$ in 1.5% $K_4Fe(CN)_6$ and 0.1 M sodium cacodylate, en bloc stained with 0.5% uranyl magnesium acetate, dehydrated and embedded in Embed 812. Electron microscopy reagents were purchased from Electron Microscopy Sciences (Hatfield, PA). For morphometric analysis, cells whose entire perimeter was visible in the EM sections were selected and all clathrin-coated structures, including clathrin-coated vesicular profiles within 500 nm of the plasma membrane, were counted. The total number of coated structures per unit length of the plasma membrane of each cell was calculated and averages of 30 control and patient cells were obtained.

## Immunofluorescence

Cells were grown on 5 µg/ml fibronectin (Millipore) coated glass coverslips and fixed with 4% formaldehyde in 0.1 M sodium phosphate, pH 7.2. Coverslips were washed with 50 mM $NH_4Cl$, pH 7.2, blocked and permeabilized with PBS + 3% bovine serum albumin + 0.1% Triton X-100. Subsequent primary and secondary antibody incubations were also performed in this buffer. Coverslips were finally mounted in the Prolong Gold antifade reagent (Life Technologies, Carlsbad, CA). Samples were imaged by spinning disk confocal microscopy.

## Spinning disk confocal microscopy

Multicolor images were acquired sequentially through a 60× oil objective (1.4 NA, CFI Plan Apochromat VC) at 1-min intervals, and at least 10 frames were acquired before the addition of compound. All imaging experiments were performed on a spinning-disc confocal (SDC) microscope, using the Improvision UltraVIEW VoX system (PerkinElmer, Waltham, MA) built around a Nikon Ti-E Eclipse microscope equipped with Perfect Focus, 14-bit electron-multiplying charge-coupled device camera (C9100-50; Hamamatsu Photonics, Hamamatsu, SZK, Japan), and spinning disc-confocal scan head (CSU-X1; Yokogawa Corporation of America, Sugar Land, TX) controlled by Volocity software (PerkinElmer, Waltham, MA). Green fluorescence was excited with a 488-nm/50-mW diode laser (Coherent) and collected by a band pass (BP) 527/55-nm filter. Red fluorescence was excited with a 561-nm/50-mW diode laser (Cobolt) and collected by a BP 615/70-nm filter.

## Total internal reflection fluorescence microscopy

Total internal fluorescence microscopy was performed using a Nikon Ti-E Eclipse inverted microscope fitted with an Apo TIRF 100 × N.A. 1.49 oil objective and driven by Andor iQ software (Andor Technologies, Belfast, Ireland). Images were acquired with a back-illuminated Andor iXon 897 EMCCD camera (512 × 512, 14 bit; Andor Technologies). Cells were imaged by sequential excitation at 0.25 Hz. Post-acquisition image analysis was performed in ImageJ software (1.46j, NIH) and GraphPad Prism.

## Image analysis

Fluorescent particle detection, lifetime tracking, and lifetime analysis of clathrin-coated pits in control and patient cells expressing GFP-tagged µ2 subunit of AP2 (µ2-GFP) was performed as previously described (*Liang et al., 2014*). Patient cells were rescued by expressing mCherry-OCRL[WT]. For this analysis all images were acquired using TIRF microscopy. 12 control cells, 12 patient cells and 7 rescued patient cells were used and an average of 809 events per cell were analyzed. Only endocytic pits that

appeared during the acquisition and with a lifetime of at least 12 s were selected. An automated tracking analysis was used to determine the lifetime distributions of µ2 puncta for each cell. Using expectation-maximization algorithms, a Gaussian mixture model was then fitted to each lifetime histogram independently, identifying three subpopulations (short, medium and long lifetimes). The mean lifetimes of the three subpopulations across different cells were calculated (40 s, 108 s, and 244 s respectively), fixed and then Gaussian models using these means for each subpopulation were re-fitted to the histograms.

Particle counting was conducted in ImageJ (National Institutes Of Health, Bethesda, MD) by thresholding and using the analyze particles plug-in. To examine the sequential recruitment of proteins at the pit, raw images were analyzed with ImageJ to generate intensity plots of areas of interest and to quantitate particles as described previously (*Perera et al., 2006*; *Zoncu et al., 2007*). Briefly, average fluorescence/time plots were generated from individual fluorescent spots, which were subsequently time-aligned by the conversion of one fluorescent marker to another. For each time point, the fluorescence intensity was normalized to the peak fluorescence intensity of that pit. For these analyses, clathrin-coated pits were manually selected at random. Statistical significance was calculated by two-tailed Student's *t* test.

## Transferrin uptake

Uptake of biotinylated transferrin was performed as previously described (*Yarar, et al., 2005*) with slight modifications. Briefly, control and patient cells were starved for 1.5 hs, then chilled on ice for 30 min and finally incubated with biotinylated transferrin (10 µg/ml) in ice-cold DMEM on ice for 45 min. Cells were then washed with cold PBS and incubated with pre-warmed culture media at 37°C for the times indicated. Internalization was stopped by placing the cells on ice and washing them three times with cold PBS. Cells were then incubated on ice with avidin (0.05 mg/ml) for 1 hr followed by incubation with biocytin (0.05 mg/ml) for 15 min. Cells were then washed three times with PBS and lysed (1% TX-100, 0.1% SDS, 0.2% BSA, 50 mM NaCl, 1 mM Tris, pH 7.4). Cell lysates were then added to ELISA plates coated with anti-human transferrin antibody (Abcam) and assayed for detectable biotinylated transferrin using chromogen-conjugated streptavidin as indicated in the manufacturer's protocol. In *Figure 4K* internalized biotinylated transferrin was expressed as the percent of total surface-bound at 4°C, which was not incubated with avidin or biocytin. Uptake of fluorescently label transferrin was performed in a very similar fashion, as previously described (*Ritter et al., 2013*). Cells were incubated with transferrin-Alex 594 instead, placed on ice after internalization, washed three times with cold PBS, and surface bound transferrin was removed by a quick acid wash (0.2 M acetic acid, 0.5 M NaCl). Cells were then fixed with 4% PFA and imaged by spinning disk confocal microscopy.

## Transferrin receptor biotinylation

Cell were rinsed with PBS, labeled on ice for 60 min with 1 mg/ml EZ-link Sulfo-NHS- SS-Biotin (Thermo Scientific, Rockford, IL), rinsed with PBS and lysed in PBS containing 1% TX100 and 0.1%SDS and protease inhibitor cocktail (Roche, Indianapolis, IN). Biotinylated proteins were recovered on neutravidin beads (Thermo Scientific) and eluted by reduction with 2-mercaptoethanol containing SDS-PAGE sample buffer. Evaluation of transferrin receptor levels in starting material, biotinylated (cell surface) and non-biotinylated (intracellular) fractions was assessed by immunoblotting following SDS-PAGE. Actin immunoblotting (mouse anti-actin antibody; Sigma) was used as a negative control to ensure that the assay specifically distinguished between cell surface and intracellular proteins.

## Subcellular fractionation

Particulate and cytosolic fractions were purified from control and patient cells using ice-cold buffer A (100 mM MES, pH 6.5, 1 mM EGTA, and 0.5 mM $MgCl_2$). Cells were harvested, pelleted, washed in PBS and repelleted. The cell pellet was resuspended in buffer A, homogenized in a glass Teflon homogenizer, passed through a 25G, 5/8 in needle and centrifuged for 5 min at 800×*g* in a tabletop centrifuge at 4°C. The supernatant was collected and centrifuged for 1 hr at 60,000×*g* in a TLA 100.2 rotor at 4°C. The resulting supernatant (cytosol) and pellet (particulate fraction) were analyzed by SDS-PAGE and Western blotting.

## Protein purification

Recombinant NH2-terminal OCRL fragments were expressed in *Escherichia coli* BL21 as GST-tagged fusions as previously described (*Mao et al., 2009*). Fusion proteins were purified on glutathione Sepharose beads according to standard protocols.

## Pull-downs and co-immunoprecipitations

Adult mouse brain extracts were prepared by homogenization in lysis buffer (PBS, 0.5% Triton [vol/vol], protease inhibitor mixture [Roche]) followed by ultracentrifugation (100,000×$g$, 60 min, 4°C) to remove insoluble material. Cleared lysates were then incubated with GST-fusion proteins on beads for 2 hr at 4°C. After extensive wash, bound proteins were analyzed by SDS–PAGE.

For co-immunoprecipitations, HEK 293 cells transiently expressing GFP-tagged constructs or HeLa cells stably expressing GFP alone or GFP-OCRL were collected and lysed in lysis buffer (150 mM NaCl, 10 mM Tris, pH 7.5, 1% Triton X-100, 1 mM EDTA, 200 U benzonase [Merck, Darmstadt, Germany] and cOmplete protease inhibitor cocktail tablet [Roche Diagnostics, Indianapolis, IN]). Cell extracts were incubated on ice for 15 min followed by centrifugation at 4000×$g$ and 4°C for 15 min. Cleared lysates (2–4 mg) were incubated under rotation for 30 min at 4°C with 50 μl magnetic beads or 25 μl sepharose beads coupled to monoclonal mouse anti-GFP antibody (Miltenyi Biotec Inc., Auburn, CA). Following centrifugation and several washes in lysis buffer, proteins recovered on beads were eluted with SDS sample buffer and separated by SDS-PAGE.

## Phosphoinositide analysis

Analysis of phosphoinositides was performed by HPLC separation of glycerol-inositol phosphates after phospholipid deacylation and analysis of the mono- and bis-phosphate species by conductivity detection (*Nasuhoglu et al., 2002*). Given that the bulk of cellular PIP2 and PIP are represented by PI(4,5)P2 and PI(4)P respectively, the two peaks closely reflect the levels of these two phosphoinositides.

## Acknowledgements

We are thankful to Frank Wilson, Louise Lucast and Lijuan Li for superb technical assistance, to Abel Alcazár-Román, Olof Idevall-Hagren and Christina Whiteus for technical consultation, to Shawn Ferguson and Hongying Shen for helpful discussions and critical reading of the manuscript, to Michelle Pirruccello and Michael Caplan for advice, to Dr Wen-Hann Tan (Boston Children's Hospital, Boston, MA) and Robert Nussbaum (UCSF) for making available de-identified Lowe syndrome patient fibroblasts. Many generous gifts of key reagents are acknowledged in the 'Materials and methods' section. This work was supported in part by NIH grants DK082700, R37NS036251 and DK45735 to PDC, NIH training grant MSTP TG T32GM07205 to DMB, by grants from the Lowe Syndrome Association, the Lowe Syndrome Trust and the Ellison Medical Foundation to PDC, and by a grant from the W.M. Keck foundation to PDC and JSD. MYH and MM were supported by the German Federal Ministry of Education and Research (FKZ01GS0861, DiGtoP consortium).

## Additional information

### Funding

| Funder | Grant reference number | Author |
| --- | --- | --- |
| National Institutes of Health | Medical Science Training Program Training Grant T32GM07205 | Daniel M Balkin |
| National Institute of Diabetes and Digestive and Kidney Diseases | DK082700 | Pietro De Camilli |
| Lowe Syndrome Association | | Pietro De Camilli |
| Lowe Syndrome Trust | | Pietro De Camilli |
| German Federal Ministry of Education and Research | FKZ01GS0861, DiGtoP consortium | Matthias Mann |
| National Institute of Diabetes and Digestive and Kidney Diseases | DK045735 | Pietro De Camilli |
| National Institute of Neurological Disorders and Stroke | R37NS036251 | Pietro De Camilli |
| Howard Hughes Medical Institute | | Pietro De Camilli |

Cell biology | Human biology and medicine

| Funder | Grant reference number | Author |
|---|---|---|
| W. M. Keck Foundation | | James S Duncan, Pietro De Camilli |

The funders had no role in study design, data collection and interpretation, or the decision to submit the work for publication.

## Author contributions

RN, DMB, MM, MYH, Conception and design, Acquisition of data, Analysis and interpretation of data, Drafting or revising the article; LL, MM, PDC, Conception and design, Analysis and interpretation of data, Drafting or revising the article; SP, HC, Acquisition of data, Analysis and interpretation of data; JSD, Conception and design, Drafting or revising the article

## Author ORCIDs

Marco Y Hein, http://orcid.org/0000-0002-9490-2261

## Additional files

### Supplementary file

• Supplementary file 1. List of OCRL interaction partners recovered through quantitative label-free mass spectrometry proteomics. (**A**) Specific interaction partners of OCRL from label-free pulldowns of HeLaM-GFP vs HeLaM-GFP-OCRL near endogenous levels. (**B**) Specific interaction partners of OCRL from label-free pulldowns of HeLaM-GFP vs HeLaM-GFP-OCRL at 5X endogenous levels.

### Major dataset

The following dataset was generated:

| Author(s) | Year | Dataset title | Dataset ID and/or URL | Database, license, and accessibility information |
|---|---|---|---|---|
| Nandez R, Balkin DM, Messa M, Paradise S, Czapla H, Hein MY, Mann M, De Camilli P | 2014 | Data from: A ROLE OF OCRL IN CLATHRIN-COATED PIT FISSION AND UNCOATING REVEALED BY STUDIES OF LOWE SYNDROME CELLS | doi:10.5061/dryad.n5p7g | Available at Dryad Digital Repository under a CC0 Public Domain Dedication. |

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
