## [Decision Letter]

Thank you for sending your work entitled “A role of OCRL in clathrin-coated pit fission and uncoating revealed by studies of Lowe syndrome cells” for consideration at *eLife.* Your article has been favorably evaluated by Randy Schekman (Senior editor) and 3 reviewers, one of whom, Suzanne Pfeffer, is a member of the Board of Reviewing Editors.

The Reviewing editor and the other reviewers discussed their comments before we reached this decision, and I assembled the following comments to help you prepare a revised submission.

This is a well carried out study that appears to show, for the first time, a role for OCRL protein in late stages of clathrin coated vesicle pit fission and uncoating. Using cells from patients with Lowe's syndrome, the authors find accumulations of clathrin and adaptor proteins and Vps9 under the plasma membrane. These structures nucleate actin comets not seen in wild type cells. The findings reported here potentially fill an important conceptual gap in the endocytosis field as it has remained unclear how and at which point in the pathway PI(4,5)P2 hydrolysis occurs in non-neuronal cells that express very low levels of synaptojanin 5-phosphatases. The work will be of broad interest with modifications as described here.

1) Loss of OCRL causes a defect in CCP maturation at the level of u-shaped CCPs that apparently fail to constrict, yet the numbers of CCVs seen in electron micrographs is normal, a finding that seems inconsistent with the accumulation of clathrin-coated endocytic vesicles propelled by actin comets in OCRL mutant cells. Please clarify. What is the fraction of actin comets harboring clathrin at their tips? Is this seen for endogenous proteins; i.e. could the clathrin-coated tips observed at actin comets be resulting from clathrin light chain overexpression used to visualize these structures?

2) The authors favor a model in which trapping of endocytic proteins in CCVs that fail to uncoat in absence of OCRL-mediated PI(4,5)P2 hydrolysis indirectly causes stalling of CCP dynamics and an accumulation of u-shaped CCPs. The evidence in favor of this model is by TIRF imaging that OCRL appears to be recruited late to endocytic structures, in fact, at a time point when clathrin uncoating sets in. However, it is also possible as the authors state, that a fraction of OCRL acts early in the pathway. Is the primary role of OCRL is in CCP maturation or in uncoating? The authors seem to prefer a model in which SNX9 acts prior to OCRL and potentially recruits OCRL to late-stage CCPs but do not provide data to support this. If OCRL is recruited to CCPs after the peak of clathrin recruitment then the majority of CCPs should lack detectable OCRL signals when imaged by confocal microscopy. Please quantify this. Conversely, substantial colocalization of OCRL with plasma membrane CCPs might be considered a sign of an early role of OCRL in addition to the late pool observed in TIRF imaging.

3) Regarding the mechanism of OCRL action and its relationship to SNX9: Does depletion of SNX9 (and/ or related SNXs) perturb OCRL recruitment to CCPs? What is the role of SNX9, which accumulates at CCPs and is overexpressed in OCRL mutants, in actin comet tail generation? If KD of SNX9 in OCRL mutants would abolish actin comets this would be a strong indication for a close functional and physical link between OCRL-mediated PI(4,5)P2 turnover and SNX9 regulation of actin polymerization.

4) Other studies of patient fibroblasts have seen defects in internalization of certain, but not all cargoes. The authors use a fluorescent assay to monitor uptake and a single time point, surface biotinylation assay. Since other labs seem to have missed this defect, and Tf internalization should continue linearly for ∼20 minutes, it seems to this reviewer that a biotin uptake assay continuing on the time zero findings of Figure 4 would strengthen the story and cement their conclusions in relation to previous work from other labs; please provide a quantitative assay of endocytosis that extends to 20 minutes.

5) The number of CCPs analyzed in Figures 2 and 4 is very low and ought to be increased to derive valid conclusions.

6) Not essential but please comment: The SNX9 subfamily includes other members such as SNX18 and SNX33. Do these bind to OCRL as well and accumulate at CCPs in OCRL mutant fibroblasts? Does KD of PIPK isoforms rescue stalled CCP dynamics in cells lacking OCRL? Data in this direction would further strengthen the link between OCRL and PI(4,5)P2 levels and their effect on CCP maturation.

---

## [Author Response]

We were very pleased by the positive assessment of our study and we thank the reviewers for their helpful suggestions. We have addressed each of the points raised and modified our manuscript accordingly.

Major changes include a new quantitative, biochemical uptake assay, the impact of SNX9 knockdown on the comets, an increase of EM samples analyzed showing the buildup of coated vesicles as well as wide-neck pits, and the automated analysis of clathrin-coated pit dynamics based on thousands of events. New and edited figures include Figure 2, Figure 2—figure supplement 1, Figure 4, Figure 5, Figure 5—figure supplement 1 and Figure 5—figure supplement 2

We have slightly modified the text to better reflect our interpretation that endocytic proteins are sequestered on uncoated vesicles in the absence of OCRL leading to a defect in clathrin-coated pit dynamics and endocytosis. Accordingly, we made a minor modification to the original title and Abstract.

*1) Loss of OCRL causes a defect in CCP maturation at the level of u-shaped CCPs that apparently fail to constrict, yet the numbers of CCVs seen in electron micrographs is normal, a finding that seems inconsistent with the accumulation of clathrin-coated endocytic vesicles propelled by actin comets in OCRL mutant cells*.

Please clarify. What is the fraction of actin comets harboring clathrin at their tips? Is this seen for endogenous proteins; i.e. could the clathrin-coated tips observed at actin comets be resulting from clathrin light chain overexpression used to visualize these structures?

The number of CCVs is indeed higher in patient cells. This was already shown in the original submission. Given the importance of this point, we have now doubled the number of cells analyzed by EM and corroborated evidence for a robust, statistically significant increase in the number of CCVs in patient cells. We have modified Figure 4 to reflect our newest analysis. For both the original figure and the new figure we have only analyzed CCVs in proximity of the plasma membrane (within 500 nm), as CCVs located more deeply into cells may also represent Golgi-derived vesicles. Since CCVs rapidly disperse throughout the cell upon fission, our data may actually represent an underestimate of the total number of endocytic CCVs that accumulate in Lowe syndrome patient cells.

Concerning the fraction of actin comets harboring endocytic clathrin coat proteins at their tip, we measured this fraction for clathrin, AP-2 and SNX9. We found that nearly 100% of the comets are positive for these proteins at their head and have included this analysis in Figure 5. The presence of these proteins at comet tips was confirmed by immunofluorescence of the endogeneous proteins. Examples of endogenous clathrin and SNX9 at the tip of comets are now shown in Figure 5.

*2) The authors favor a model in which trapping of endocytic proteins in CCVs that fail to uncoat in absence of OCRL-mediated PI(4,5)P2 hydrolysis indirectly causes stalling of CCP dynamics and an accumulation of u-shaped CCPs. The evidence in favor of this model is by TIRF imaging that OCRL appears to be recruited late to endocytic structures, in fact, at a time point when clathrin uncoating sets in. However, it is also possible as the authors state, that a fraction of OCRL acts early in the pathway. Is the primary role of OCRL is in CCP maturation or in uncoating? The authors seem to prefer a model in which SNX9 acts prior to OCRL and potentially recruits OCRL to late-stage CCPs but do not provide data to support this. If OCRL is recruited to CCPs after the peak of clathrin recruitment then the majority of CCPs should lack detectable OCRL signals when imaged by confocal microscopy. Please quantify this. Conversely, substantial colocalization of OCRL with plasma membrane CCPs might be considered a sign of an early role of OCRL in addition to the late pool observed in TIRF imaging*.

Yes, based on our results, we strongly favor a model in which a primary function of OCRL is at the uncoating stage. As we show in this manuscript and as we have previously shown (24), OCRL is recruited only at very late stages at CCPs. Additionally, as we mention in the discussion of the manuscript, a previous study by Taylor et al. had also shown a sharp recruitment of OCRL at the very end of the CCP lifetime. Interestingly, this study also showed that the OCRL recruitment profile precisely coincided with that of GAK, a key clathrin uncoating factor ([73], Figure 4). We note that, similar to our current results, lack of GAK results in an impairment of endocytosis due to sequestration of clathrin in assembled coats (Lee *et al.*, 2005, Yim *et al.*, 2010). In response to this comment of the referee, we have modified the text by emphasizing how the partial stalling of clathrin coated pit maturation at mid-stage (U-shaped pits) may only be an indirect effect of the sequestration of endocytic factors on uncoated vesicles. We have calculated the fraction of CCPs positive for OCRL at any given time and found that such fraction is below 25 % in control fibroblasts.

*3) Regarding the mechanism of OCRL action and its relationship to SNX9: Does depletion of SNX9 (and/ or related SNXs) perturb OCRL recruitment to CCPs? What is the role of SNX9, which accumulates at CCPs and is overexpressed in OCRL mutants, in actin comet tail generation? If KD of SNX9 in OCRL mutants would abolish actin comets this would be a strong indication for a close functional and physical link between OCRL-mediated PI(4,5)P2 turnover and SNX9 regulation of actin polymerization*.

In response to this comment we performed KD experiments of SNX9 and of its homologue SNX18. Microscopy observations suggested a decrease in the number of comets but with variability from cell to cell, which roughly correlated with remaining levels of SNX9 immunoreactivity. Interestingly, even in KD cells, when comets were observed they had tips positive for SNX9 immunoreactivity. These results are consistent with a role of SNX9 in the actin polymerization leading to actin tails. These data are now shown in Figure 5—figure supplement 2. We note however, that interaction of OCRL with SNX9 is not required for its recruitment to clathrin coated pits, as an OCRL mutant harboring a mutation in the SNX9 binding site, but still capable of binding clathrin and AP-2, was still recruited to very late stage clathrin coated pits (Figure 5—figure supplement 2).

*4) Other studies of patient fibroblasts have seen defects in internalization of certain, but not all cargoes. The authors use a fluorescent assay to monitor uptake and a single time point, surface biotinylation assay. Since other labs seem to have missed this defect, and Tf internalization should continue linearly for ∼20 minutes, it seems to this reviewer that a biotin uptake assay continuing on the time zero findings of*
Figure 4
*would strengthen the story and cement their conclusions in relation to previous work from other labs; please provide a quantitative assay of endocytosis that extends to 20 minutes*.

As requested, we have performed a quantitative, 20 minutes time-course assay of endocytosis by monitoring the uptake of biotinylated transferrin by ELISA (protocol adapted from S. Schmid’s lab; Yarar *et al.,* 2005). The results of these experiments confirm the observations reported in our original manuscript, as they show a delay in endocytosis in Lowe syndrome patient cells compared to controls. These new data are now shown in Figure 4, replacing the fluorescent transferrin time-course uptake data.

*5) The number of CCPs analyzed in*
Figures 2 and 4
*is very low and ought to be increased to derive valid conclusions*.

In order to increase the number of CCPs analyzed, we have used a recently published automated tracking method (41) that allows the calculation of the lifetime distributions of CCPs. We have examined the lifetime distributions of over 6500 events per condition, increasing our sample size over 400-fold. The results of this analysis, shown in Figure 4, are consistent with our previous observations indicating that the lack of OCRL in patient cells leads to a delayed turnover and extended CCPs lifetime.

For Figure 2, which focuses on the sequential recruitment of SNX9 and OCRL to the pit, we increased the number of events manually scored by 10-fold. This improved analysis is shown in Figure 2, replacing the previous fluorescence traces.

*6) Not essential but please comment: The SNX9 subfamily includes other members such as SNX18 and SNX33. Do these bind to OCRL as well and accumulate at CCPs in OCRL mutant fibroblasts? Does KD of PIPK isoforms rescue stalled CCP dynamics in cells lacking OCRL? Data in this direction would further strengthen the link between OCRL and PI(4,5)P2 levels and their effect on CCP maturation*.

We tested an overexpressed GFP-SNX18 construct and observed that it accumulates at the tip of actin comets in patient cells, akin to SNX9 (Figure 5). In addition, we observed that SNX18 is also capable of co-immunoprecipitating OCRL and we have included these data in Figure 2—figure supplement 1. The expression levels of SNX33 are very low in our patient and control cells as assayed by qPCR, and we decided not to focus on this protein.

We have not investigated the effects of knocking down PIPKs on CCP dynamics in patient cells. Given the occurrence of different isoforms and their potential functional overlap, we expect this analysis to lead to inconclusive results.